# Effective and Robust Log-Linear Policy Design in an Economy-SIR Model using Two-Level Reinforcement Learning

## Abstract

Optimizing economic policies in the face of severe economic shocks, such as pandemics, poses a complex multi-agent challenge. Here we show that the AI Economist framework can learn sets of effective, robust, and explainable policies using two-level reinforcement learning (RL) and data-driven simulations. Our framework can optimize for a wide range of (mis)aligned social welfare objectives and accounts for strategic behavioral responses. In contrast, existing analytical and computational methods do not scale to this setting. We validate our framework on optimizing state policies and federal subsidies in an unemployment-vaccination-SIR simulation fitted to US COVID-19 data. We find that log-linear RL policies significantly improve public health and economic metrics compared to real-world outcomes. In particular, federal subsidies can align incentives more between state and federal agents. Their behavior is explainable, e.g., RL policies respond strongly to changes in recovery and vaccination rates. They are also robust to calibration errors, e.g., infection rates that are over or underestimated. As of yet, real-world policymaking has not seen adoption of machine learning methods at large, including RL and AI-driven simulations. Our results show the potential of AI to guide policy design and improve social welfare amidst the complexity of the real world.

## 1 Introduction

Designing policies to address urgent socioeconomic challenges poses an intricate multi-agent optimization problem. Real-world policies should be effective, robust, explainable, and implementable. Furthermore, policy making often takes place across different jurisdictions and levels of government. We focus on a hierarchical setting with multiple levels of policy makers (*agents*), e.g., state and federal governments, that need to respond to shocks to society, e.g., the COVID-19 pandemic.[1] Following standard economics terminology and for clarity, we will refer to the highest-level agent as the "planner". Each agent sets policy for its jurisdiction, which may or may not overlap, e.g., country vs states. Each agent is assumed strategic and needs to consider trade-offs between multiple policy objectives and levers.

Few tractable analytical or computational solutions exist for this well-motivated, but complex multi-agent policy design problem. The agents may have misaligned objectives, which may discourage cooperation and decrease overall social welfare. The planner may face a *mechanism design* problem (Myerson, 1981), as its actions may change the incentives and objectives of other agents. For instance, federal subsidies can boost people's income, prevent bankruptcies, and protect a state's economy, whose objective could be to maintain economic productivity. However, optimal mechanisms and their equilibria are generally hard to find, especially in the dynamic setting with economic shocks and when all agents are strategic.

---

[1]Where unclear, we explicitly use "US" in "US state" to disambiguate it from "state" as used in machine learning. The term "policy" can refer to the decisions made by an agent, as in "public policy", or to the behavioral model of a reinforcement learning agent.

**Two-Level RL for Policy Design.** This work builds on the AI Economist framework to learn effective, robust, and explainable policies in this hierarchical and dynamic setting. The AI Economist combines *two-level* (or "bi-level"), multi-agent reinforcement learning (RL) with economic simulations grounded in real-world data. This framework was previously used to design AI income tax policies that improve equality and productivity (Zheng et al., 2021).

The AI Economist framework fills an important methodological gap in the space of policy design. Existing analytical approaches are limited to a narrow set of policy objectives or do not yield explicit solutions in the dynamic, multi-timestep setting. Empirical methods lack counterfactual data to estimate behavioral responses (the *Lucas critique*). Moreover, existing simulation-based approaches often do not consider all agents to be strategic, or interactions between agents (Fernández-Villaverde; Benzell et al.; Hill et al., 2021). In contrast, the AI Economist uses multi-agent RL to find policies for all agents that perform well empirically, while simulations enable counterfactual analysis.

**Designing Pandemic Response Policies.** We validate our framework through optimizing public and economic policies in response to the economic shock of a pandemic, maximizing both public health (e.g., reducing deaths) and economic objectives (e.g., maintaining productivity). We focus on state and federal responses to COVID-19 in the US, evaluated within an integrated unemployment-vaccination-SIR simulation based on real data. Policies need to navigate a key trade-off: more stringent responses (e.g., lock-downs, business restrictions) may temper the spread of the pandemic (Flaxman et al.; Acemoglu et al.), but may lead to lower productivity, e.g., due to higher unemployment. Using real data, we simulate the spread of the disease, vaccinations, and unemployment in the United States. We then optimize the stringency level of 51 state-level response policies, including all 50 states and Washington D.C., and federal subsidies in the form of direct payments to citizens. Moreover, we constrain RL policies to be simple and hence more easily implementable, e.g., by constraining how often policy decisions change.

As a modeling assumption, the policy objective of states depends on the level of federal subsidies: higher federal subsidies may (indirectly) incentivize states to accept higher unemployment and be more stringent to reduce deaths. However, this may cost more, e.g., require borrowing money. Although the underlying public health and economic mechanisms are complex, these relationships and trade-offs are salient in real data. Hence, this presents a two-level policy optimization problem.

**Claims and Summary of Findings.** The central claims of this work are that, in a data-driven simulation of COVID-19, unemployment, and its wider economic impact in the US, two-level RL (1) can find policies that yield higher social welfare and (2) can articulate the trade-offs between welfare metrics across a range of possible policy objectives. We present the following results in support of these claims.

We find that log-linear RL policies achieve significantly higher social welfare than real-world policies executed in simulation. Across states, *well-performing agents use harder and faster response policies to lower deaths, while leading to similar levels of unemployment after about one year.* These policies perform well across a wide range of prioritizations of public health vs the economy, and demonstrate how the social planner can align state incentives with national social welfare through subsidies. The policies are explainable: well-performing policies respond strongly to changes in recovery and vaccination rates, for example. These results are also robust to a range of calibration errors, e.g., infection rates that are over or underestimated. Moreover, these policies are constrained to change slowly and hence be more implementable.

## 2 Related Work

A plethora of research, commentary, recommendations, and other work on policy design and analysis has appeared in almost real-time during the pandemic. A significant volume has been collected in thematic indices (Nature) and surveys (Padhan & Prabheesh, 2021; Brodeur et al., 2021). Below, we describe several related strands of work.

**Policy analysis.** Early work tracked policies (Hale et al., 2020) and described emerging perspectives, influential contextual and emotional factors, and the pathways of public policies that were implemented (or

not) in response and in the face of uncertainty (Weible et al., 2020). The effectiveness of policies depends on the context, e.g., government stimulus might be less effective in developing countries (Loayza & Pennings, 2020), and public acceptance of vaccines (Lazarus et al., 2021). Other early commentary has argued policies should target high-risk populations to address inequities in the impact of the pandemic (Wang et al., 2020). Public policies were compared across countries, showing enormous variations on their implied price-on-life (Balmford et al., 2020) and how they depended on the capacity to operationalize and build political support (Capano et al., 2020).

**Policy design.** Analytical work studied optimal targeted lockdown strategies with multiple age groups (Acemoglu et al., 2021), and economic policies in a dynamical economic model with a single government planner (Collard et al., 2020). Many variations of and interventions to "flatten the curve" in the SIR model have been studied (Gonzalez-Eiras & Niepelt, 2020; Toda, 2020; Bliman et al., 2021; Morris et al., 2021), including time-optimal control strategies (Bolzoni et al., 2017), the effect of treatments (Ding & Schellhorn, 2022), and optimal vaccination policies (Laguzet & Turinici, 2015). Policy design also depends on accurate forecasts, as reviewed by Rahimi et al. (2021).

However, relatively few works have studied optimal policy design in the intersection of public health and the economy with multiple levels of government policies. To the best of our knowledge, no other work has studied policy design in the intersection between unemployment, COVID-19, and policy interventions, nor has used multi-agent RL. As such, our work provides a more comprehensive analysis based on machine learning techniques compared to previous work.

**Public health aspects.** Existing public health research has focused on many aspects of COVID-19, including augmenting epidemiological models for COVID-19 (Li & Muldowney; Zou et al.), analyzing contact-tracing in agent-based simulations (Alsdurf et al.), and evaluating the efficacy of various public health policy interventions (Kapteyn et al.; Flaxman et al.; Chang et al.). An analytical approach to policy design showed that the elasticity of the fatality rate to the number of infected is a key determinant of an optimal response policy (Alvarez et al., 2020; Acemoglu et al.). Moreover, statewide stay-at-home orders had the strongest causal impact on reducing social interactions (Abouk & Heydari, 2020).

**Economic aspects.** The effects of pandemic response policies have been studied, such as the relationship between unemployment insurance and food insecurity (Raifman et al.), while various sources have tracked government expenditure during the pandemic (IRS; U.S. Department of the Treasury; Committee for a Responsible Federal Budget). Difference-in-difference analysis of location trace data finds that imposing lockdowns leads to lower overall costs to the economy than staying open (Brzezinski et al., 2020), under a modified SIR model. Early US data has also shown the unequal distribution and unemployment effects of remote work across industries (Brynjolfsson et al., 2020), which is also supported by mobile tracking data (Chang et al., 2021).

**Methodology.** Our work builds on agent-based models (Railsback & Grimm, 2019; Macal & North, 2005) and agent computational economics (Tesfatsion, 2006), which study economies as dynamic systems of individual agents and their resulting emergent phenomena. However, previous ABM models largely do not feature strategic behavioral models for the agents. In our work, we learn those behaviors through the use of deep multi-agent RL, which is now becoming technically feasible to perform with many agents.

Our multi-agent environment relates to several economic models, include general equilibrium models (Cox et al., 1985; Smets & Wouters, 2003) and models that study inter-generational inter-regional dynamics (Benzell et al.) and their resulting macro-economic, aggregate phenomena. However, these models typically do not study which equilibrium behaviors are *learned* nor interactions between individuals, while their structural estimation and calibration to real data remains challenging. From this point of view, our work demonstrates that multi-agent RL can find better-performing equilibria in well-calibrated simulations.

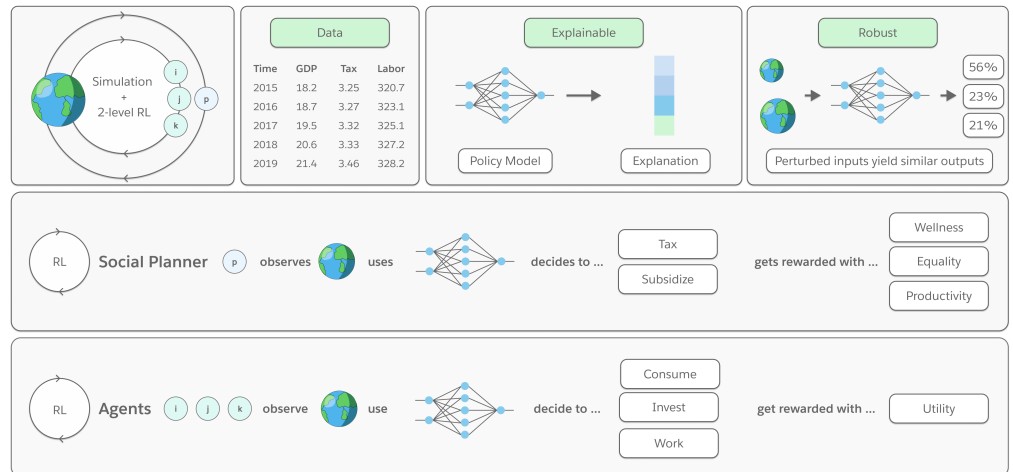

Figure 1: **The AI Economist policy design framework.** Our policy design framework uses a data-driven economic simulation with a hierarchy of strategic agents. The highest-level agent is called a planner. All agents optimize their (behavioral) policy model using RL; repeatedly observing, deciding, and updating their model using reward feedback. This poses a two-level (or *bi-level*) optimization problem, as agents strategically respond to the planner's policy. This approach yields effective, explainable, simple, and robust policies. See Section 4 for details.

## 3 Policy Design using Simulations and Two-Level RL

We describe the two elements of our policy design approach: data-driven *simulations* with AI agents, and *two-level RL*, as shown in Figure 1.

**Formal Setup.** We consider a dynamic environment with $N + 1$ agents, including a planner (denoted by $p$). We index agents with $i = 1, \ldots, N, p$. This setup can be formalized as a Markov Game (Littman, 1994). Episodes iterate for $T > 1$ time steps. In this work, each time step represents a day and episodes last for about a year. At each time $t$, each agent $i$ gets an observation $o_{i,t}$ of the environment state $s_t$, can perform actions $a_{i,t}$, and receives a scalar reward $r_{i,t}$. The environment then transitions to the next state $s_{t+1}$ using environment dynamics $\mathcal{T}(s_{t+1}|s_t, \boldsymbol{a}_t)^2$. Assuming rationality, each agent $i$ optimizes its behavioral policy $\pi_i(a_{i,t}|o_{i,t})$ to maximize its total (discounted) reward:

$$\max_{\pi_i} \mathbb{E}_{\pi_i, \boldsymbol{\pi}_{-i}} \left[ \sum_{t=0}^{T} \gamma_i^t r_{i,t} \right]. \tag{1}$$

Here, $0 < \gamma_i \leq 1$ is a discount factor emphasizing long-term ($\gamma_i \approx 1$) or short-term ($\gamma_i \approx 0$) rewards. In particular, the objective of agent $i$ depends on the behavior $\boldsymbol{\pi}_{-i}$ of the other agents (denoted by $-i$). In this work, each agent's reward at time $t$ is the marginal gain in social welfare $F_{i,t}(s_t; \alpha_i)$ in its jurisdiction: $r_{i,t} = F_{i,t+1} - F_{i,t}$. Here, $\alpha_i$ parameterizes the social welfare function. In effect, each agent optimizes its total (discounted) social welfare.

**Mechanism Design with Strategic Agents.** A key challenge of our hierarchical setting is that the planner faces a *mechanism design* problem (Myerson, 1981; Vickrey, 1961) with strategic agents. That is, the planner tries to learn the mechanism that maximizes its objective, *assuming agents respond strategically.* More concretely, the planner's actions may change the reward function of other agents, which may adapt their behavior in response.

Two-level problems appear naturally in economics and machine learning. For example, a planner that changes income tax rates changes the post-tax income that agents receive, which can change (the shape of) their

---

[2]Boldface denotes concatenation across agents, e.g., $\boldsymbol{a}_t = [a_{1,t}, \ldots, a_{N,t}, a_{p,t}]$

utility (Zheng et al., 2021). In computer vision, generative adversarial networks use two-level learning to synthesize realistic images (Goodfellow et al., 2014). In economics, Stackelberg games feature a leader that acts first and followers that respond (Von Stackelberg, 2010) to model security games, for example.

In this work, the planner's subsidies (federal level) can affect the social welfare experienced by agents (state level). The planner may attempt to align the (*a-priori* misaligned) incentives of the agents with its own, as in principal-agent problems (Grossman & Hart, 1992). Moreover, the two-level setting is distinct from games with a fixed reward function in which deep RL has reached (super)-human skill, e.g., Go (Silver et al., 2017) and Starcraft (Vinyals et al., 2019).

**Two-Level RL.** In this work, we use RL to both optimize the planner's mechanism and the agents' policies, given a mechanism. This is a *two-level learning problem* which is highly unstable, as agents need to readjust their behavior every time the planner changes the mechanism.[3] More generally, MARL with fixed reward functions is challenging because each agent faces a non-stationary environment when other agents are also learning and changing their behavior (Buşoniu et al., 2010). Our two-level, multi-agent RL solution builds on intuitive curriculum strategies that have been effective in other settings (Zheng et al., 2021).

In our setting, learning the best agent responses for a given mechanism amounts to finding a general-sum equilibrium in a dynamic game. However, few constructive solutions or theoretical guarantees to find or enumerate such equilibria exist in general. We empirically find that our solution can consistently find well-performing policies that achieve higher social welfare than baselines. However, there are no general theoretical guarantees on how close RL policies are to true equilibria, nor to the optimal mechanism.

**Flexible Policy Objectives.** A key benefit of our framework is that RL can optimize for any quantitative objective, even non-analytical or non-differentiable ones. This makes it very flexible and easier to align with real-world objectives, such as health, productivity, equality, or combinations thereof. In contrast, classic economic methods often use abstract objectives, such as a planner optimizing total utility (Mas-Colell et al., 1995). Such objectives can offer theoretical guarantees and analytical tractability but might be unrealistic. For instance, it is hard to measure utility in practice, or human behavior might not optimize a stylistic utility function.

## 4 Simulation and Policy Design

We now apply the AI Economist framework to optimize state response policy and federal subsidies in a simulated COVID-19 pandemic. We simulate the impact of policy choices on overall economic output and the spread of the disease. Our modeling choices are driven by what can be reasonably calibrated from publicly available health and economic data from 2020 through mid-2021. Future simulations could be expanded with the availability of more fine-grained data. This section provides an overview of the simulation design and calibration; more details are provided in the Methods (Sections 4.2-9.3).

### 4.1 Assumptions

The available data do not support modeling all aspects of the COVID-19 pandemic. We outline three salient ones below.

**Modeling assumption: Recovered people do not become susceptible again.** In reality, there have been people who have been infected more than once. This feature could technically be included in the SIR model.

However, recovered individuals have a much lower chance of contracting and spreading the infection. At the time of writing and to the best of our knowledge, there is little data on reinfections. Hence, inferring the effects on the SIR parameters or dynamics by simulating reinfections would rely on an estimate of the

---

[3]Few tractable analytical solutions exist for mechanism design, even without agents that are strategic. For example, there is no full analytical solution for auctions with two items and multiple bidders (Daskalakis, 2015).

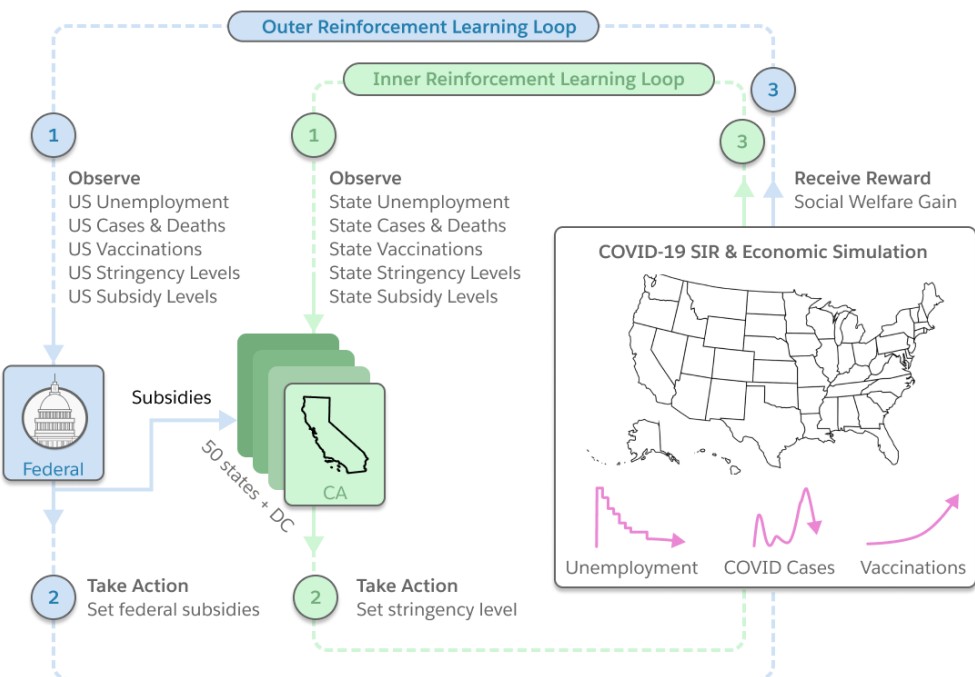

Figure 2: **Policy design using an unemployment-vaccination-SIR model and two-level reinforcement learning.** We learn policies for a planner (federal government) and agents (states). Each agent takes actions using its policy model, based on an observation of the simulation. All agents are rewarded by increases in social welfare *in their jurisdiction*. As a modeling choice, federal subsidies can offset the economic impact for states, which may incentivize states to respond more stringently to the pandemic. As such, this poses a two-level RL problem in which states optimize their policy, partially in response to federal policy.

likelihood of reinfections, getting sick, and being tested positive. This is almost impossible to do giving a lack of reliable data, and hence has been omitted in this work.

**Modeling assumption: No infection spread between states.** In our multi-region SIR model, the states do not spread infections across their borders. However, our model already fits the available data very well. For example, Figure 3, shows that the discrepancy in US deaths between the red and black lines are at most 5-10% for at most 25% of the time (period between 250-350 days since 3/22/2020). Moreover, detailed data on movement across states is also not readily available. As such, we omitted cross-region infection effects.

**Modeling assumption: Subsidies are distributed similarly across US states.** In our model, each state receives subsidies exactly proportional to their population relative to the whole US. In reality, the level of subsidies depended on income, and income inequality across states means that possibly a larger fraction of the population in some states received COVID-19 subsidy checks.

In our model, a state who receives more subsidy may be more willing to close down more. However, to the best of our knowledge, there is no reliable data on how payment checks were distributed per state, while modeling income distributions, tax-filing status, etc, would be challenging. However, at first-order, we find that disparate subsidy levels across states are likely a second-order effect. Our experiments show that the federal RL agent spends significantly less on subsidies. Hence, discrepancies between statesubsidy levels are not expected to meaningfully change our high-level conclusions.

## 4.2 Notation and Definitions

We consider two types of agents:

- $N = 51$ state governors, indexed by $i \in 1, \ldots, N$, including Washington D.C., and

- the federal government, indexed by $i = p$.

Each state $i$ has a population of $n_i$ people. The US has a population of $\sum_i n_i$ people. Agents are indexed by $i = 1, \ldots, N, p$. For a full overview of used variables, see Table 1 and Table 2 in the Supplementary Material.

## 4.3 Simulation Dynamics

Our simulation aims to capture the impact of public health policy on the spread of COVID-19 and on the economy across the US. Based on available data, we model (separately for each state) the spread of COVID-19 using a SIR model (Kermack et al.) and model economic output via unemployment and direct payments provided by the federal government.

### 4.3.1 Epidemiology Model

We use an augmented SIR-model (Kermack et al.) to model the evolution of the pandemic, including the effect of vaccines. We only model the outbreak of a single variant of COVID-19. The standard SIR model emulates how susceptible individuals can become infected and then recover. By convention, recoveries include deaths. As a simplifying assumption, only susceptible individuals are vaccinated, recovered individuals cannot get reinfected, and vaccinated individuals gain full immunity to COVID-19, directly moving from susceptible to recovered.[4] Within each state, the infection rate (susceptible-to-infected) is modeled as a linear function of the state's stringency level. This reflects the intuition that imposing stringent public health measures can temper infection rates. This relationship is also salient in real-world data.

Mathematically, the SIR model subdivides a population $n_i$ into 3 groups: susceptible $S_{i,t}$, infected $I_{i,t}$, and recovered $R_{i,t}$. The subscript $i \in \{1, \ldots, N\}$ denotes a state. The subscript $t$ denotes time, where each timestep represents a single day. Susceptible individuals become infected (transition from $S$ to $I$) at a rate that depends on the number of infected individuals and the (policy-dependent) transmission rate $\beta_{i,t}$. Infected individuals "recover" (transition from $I$ to $R$) at a rate of $\nu$, which is the same across all states. By convention, $R$ includes deaths $D_{i,t}$: "recovered" individuals die at a rate of $\mu$, also the same across all states.

To model vaccinations, we add an additional vaccinated group $V_{i,t}$. Susceptible individuals that receive a vaccine transition directly from $S$ to $V$ and therefore cannot contract, spread, or die from the disease. This is a simplifying assumption, although it is straightforward to extend this model to the case where vaccinated people have a (much) lower chance of contracting and spreading the disease.

The SIR and vaccination dynamics are captured in the following equations at the state-level:

$$S_{i,t} = S_{i,t-1} - \beta_{i,t} \frac{S_{i,t-1} I_{i,t-1}}{n_i} - \Delta V_{i,t} \tag{2}$$

$$I_{i,t} = I_{i,t-1} + \beta_{i,t} \frac{S_{i,t-1} I_{i,t-1}}{n_i} - \nu I_{i,t-1} \tag{3}$$

$$R_{i,t} = R_{i,t-1} + \nu I_{i,t-1} \tag{4}$$

$$V_{i,t} = V_{i,t-1} + \Delta V_{i,t} \tag{5}$$

$$D_{i,t} = \mu \cdot R_{i,t} \tag{6}$$

$$n_i = S_{i,t} + I_{i,t} + R_{i,t} + V_{i,t}. \tag{7}$$

Note that the transmission rate $\beta_{i,t}$ is location- and time-specific; this reflects differences between states and the stringency level of each state's public health policy (which can vary over time). Furthermore, following

---

[4]This simplifying assumption is motivated by empirical results that vaccines commonly used in the US are up to 95% effective (Pilishvili et al., 2021).

the standard SIR model, we do not include population growth: the population $n_i$ is fixed and does not depend on time. Moreover, we do not model infections between states, e.g., infection through neighbors or airplane routes.

Note that our SIR-vaccine model is a natural extension where vaccinated people directly transition from $S$ to $V$, and therefore, cannot contract, spread or die from the disease. This extension has a closed-form solution using standard mathematical methods for differential equations for $S(t)$, $I(t)$, etc, similarly to how the standard SIR model can be solved, as vaccinations grow linearly over time (Kopfová et al., 2021).

We model the transmission rate of state $i$ at time $t$ as a linear function of the State's stringency level $\pi_{i,t-d}$, delayed by $d$ days:

$$\beta_{i,t} = \beta_i^\pi \cdot \pi_{i,t-d} + \beta_i^0, \tag{8}$$

where $\beta_i^\pi$ and $\beta_i^0$ are the slope and intercept of the state-specific linear function.

$\Delta V_{i,t}$ denotes new daily vaccinations. The onset of vaccines is delayed until the simulation has reached a specified date $T_0$. After that date, a fixed amount of vaccines are dispersed daily to each state, that amount being proportional to the population size $n_i$, at a rate $v_i$. Specifically:

$$\Delta V_{i,t} = \Delta V_i \cdot \mathbf{1}[t \geq T_0], \quad \Delta V_i = v_i n_i, \quad 0 < v_i < 1, \tag{9}$$

where $\mathbf{1}$ is the indicator function.

### 4.3.2 Economic Model

We now describe the economic aspects we model that are directly relevant to the impact of COVID-19.

For each state, daily economic output is modeled as the sum of incoming federal subsidies plus the net production of actively employed individuals. At the federal level, this output is taken as the sum of the state-level outputs, minus borrowing costs used to fund the outgoing subsidies.

We model *unemployment* using a time-series model that predicts a state's unemployment rate based on its history of stringency level increases and decreases.[5] The daily productivity per employed individual is calibrated such that yearly GDP at pre-pandemic employment levels is equal to that of the US in 2019.

Unemployment in state $i$ at time $t$ is modeled by convolving the history of stringency level changes with a bank of $K$ exponentially-decaying filters:

$$\tilde{U}_{i,t}^k = \sum_{t'=t-L}^{t'=t} e^{\frac{t'-t}{\lambda_k}} \cdot \Delta\pi_{i,t'}, \tag{10}$$

$$U_{i,t} = \texttt{softplus}\left(\sum_{k=1}^{K} w_{i,k} \cdot \tilde{U}_{i,t}^k\right) + U_i^0. \tag{11}$$

The term $\tilde{U}_{i,t}^k$ denotes the unemployment response captured by filter $k$, which has decay constant $\lambda_k$, $L$ is the filter length, and $\Delta\pi_{i,t}$ is the change in stringency level in state $i$ at time $t$. Each state's excess unemployment at time $t$ is computed as a linear combination of $\tilde{U}_{i,t}$ using state-specific weights $w_i$ and a $\texttt{softplus}$ function to ensure that excess unemployment is positive. Finally, unemployment $U_{i,t}$ is taken as the sum of excess unemployment and baseline unemployment $U_i^0$.

We calibrate the daily economic output per employed individual such that, at baseline levels of unemployment, total yearly GDP of the simulated US matches the actual US GDP in 2019. Each state's daily economic output $P_{i,t}$ is modeled as the total output of its working population $\omega_{i,t}$ at time $t$ plus any money provided via direct payments from the federal government at that time $\tilde{T}_{i,t}^{state}$. Concretely, the number of available workers is

$$\omega_{i,t} = \nu\left(n_i - D_{i,t} - \eta \cdot I_{i,t}\right) - U_{i,t}. \tag{12}$$

---

[5]We also account for deaths and active infections when modeling unemployment.

And daily economic output is therefore

$$P_{i,t} = \kappa \cdot \omega_{i,t} + \tilde{T}_{i,t}^{state}. \tag{13}$$

Here, $\nu = 0.6$ captures the portion of the population that is working age and $\kappa$ captures the average daily economic output per active worker. Note that individuals that have died, as well as a fraction ($\eta = 0.1$) of infected individuals, are accounted for in determining the number of available workers.

**Datasets and Calibration.** Data for the daily stringency policies are provided by the Oxford COVID-19 Government Policy Tracker (Hale et al., 2021). The date and amount of each set of direct payments issued through federal policy are taken from information provided by the COVID Money Tracker project (Committee for a Responsible Federal Budget). We use the daily cumulative COVID-19 death data provided by the COVID-19 Data Repository at Johns Hopkins University (Dong et al., 2020) to estimate daily transmission rates. Daily unemployment rates are estimated based on the monthly unemployment rates reported by the Bureau of Labor Statistics (Bureau of Labor Statistics, 2021). We fit the disease (unemployment) model to predict each state's daily transmission (unemployment) rate given its past stringency levels. We allow state-specific parameters for both models, but during fitting we regularize the state-to-state variability to model common trends across states and to prevent overfitting to noisy data.

We calibrated the simulation on data from 2020 only. We fit simulation models on data through November 2020 and use December 2020 for validation. As such, alignment between simulated outcomes during 2021 and real-world data from 2021 reflects the ability of the simulation to *forecast* policy consequences beyond the timeframe used for calibration. See Section 9.1 in the Methods for additional details regarding datasets and calibration.

### 4.4 Policies and Objectives

**Policy Levers.** We model 51 state-level policies, for all 50 states and the District of Columbia, and 1 federal policy. Each state-level policy sets a *stringency level* (between 0% and 100%), which summarizes the level of restrictions imposed (e.g., on indoor dining) in order to curb the spread of COVID-19. The stringency level at a given time reflects the number and degree of active restrictions. This definition follows the Oxford Government Response Tracker (Hale et al., 2021). The federal policy periodically sets the daily *per capita subsidy level*, or direct payment. Federal subsidies take the form of direct payments to individuals, varying from \$0 to \$55 per day per person, in 20 increments.

**Policy Constraints for Simplicity.** Intuitively, "simple" policies are more explainable and robust, and presumably more trustworthy. For example, a simple response policy does not change often or its intensity too dramatically. In this work, we constrain state policies to only change every 30 days. This parallels common economic modeling choices, such as consumption smoothing (Morduch, 1995) and sticky prices (Mankiw & Reis, 2002).

A benefit of using RL is that it is straightforward to impose policy constraints, e.g., through manually specified action masking. Empirically, standard gradient-based optimization techniques can learn well-performing policies under such constraints. More sophisticated techniques for constrained RL exist (Achiam et al., 2017). In contrast, analytical methods may struggle with constraints that are not differentiable.

**Metrics and Objectives.** The trade-off between public health and economic outcomes is salient in COVID-19 data and policy debates, although the underlying public health and economic mechanics are complex. We model the tension between these two interrelated objectives as follows.

Each agent $i$ optimizes its policy to maximize its social welfare $F_i$, which can be distinct across agents. Social welfare is defined as a weighted sum of a public health index $H_i$ and an economic index $E_i$:

$$F_i(\alpha_i) = \alpha_i H_i + (1 - \alpha_i)E_i. \tag{14}$$

Here $\alpha_i \in [0, 1]$ encodes how much agent $i$ prioritizes public health versus the economy. Each index $H_i$ (or $E_i$) is the average health (or economic) index for agent $i$ *across days*, e.g., $H_i = \frac{1}{T}\sum_{t=0}^{T} H_{i,t}$. The marginal

health index $\Delta H_{i,t}$ at time $t$ (each time-step represents a day) measures the number of new COVID-19 deaths in the jurisdiction of agent $i$. The marginal economic index $\Delta E_{i,t}$ is a concave function of total economic output (described above) at time $t$ for agent $i$.

$$\Delta E_{i,t} = \text{crra}\left(\frac{P_{i,t}}{P_i^0}, \eta\right), \quad \text{crra}(x, \eta) = \frac{x^{1-\eta} - 1}{1 - \eta}, \quad \eta > 1. \tag{15}$$

Here, $P_{i,t}$ denotes the total economic output, which includes any incoming subsidies, and $P_i^0$ is average pre-pandemic output. The federal government uses

$$P_{p,t} = \sum_{i=1}^{N} P_{i,t} - c \cdot \tilde{T}_{i,t}^{state}, \qquad P_p^0 = \sum_{i=1}^{N} P_i^0, \tag{16}$$

so that federal-level economic output accounts for the borrowing cost of funding outgoing subsidies $\tilde{T}^{state}$. The crra utility function is *concave* and a common choice to model diminishing marginal utility. Here, we use it to define $\Delta E_{i,t}$ as a function of total economic output. This captures the intuition that *a decrease in economic output is felt more severely when that output is already low.*

The form of economic index and productivity encodes the *two-level* interaction between the states and the federal government. Federal subsidies can "soften the blow" of additional unemployment, as they raise the level economic output for states. As such, subsidies indirectly incentivize additional stringency by mitigating the trade-off between $H_i$ and $E_i$ faced by states.

To simplify notation and analysis, we normalize indices so that each minimum-stringency policy yields $H_i = 0$ and $E_i = 1$ and so that each maximum-stringency policy yields $H_i = 1$ and $E_i = 0$. As a result, higher index values denote more preferable outcomes.

Finally, we infer the prioritization values $\hat{\alpha}_i$ that maximize social welfare $F_i(\hat{\alpha}_i)$ for the real-world stringency policies (see Section 9.3 in the Methods). Calibrating $\hat{\alpha}_i$ in this way gives the fairest comparison between RL and real-world policies.

### 4.5 Theoretical Considerations

**Subsidies create indirect interaction effects between states.** At first sight, the SIR and economic dynamics of the $N$ state agents are independent of each other, and may look like $N$ 2-agent problems between the federal agent and each state. However, the $N$ problems are not totally independent. Our model can be seen as $N$ weakly-coupled 2-agent problems that are coupled through the penalty on total subsidies for the federal agents. For example, the federal agent may subsidize a lot to effectively dampen infections in one state, but this also improves the economic indices for all other states. On the other hand, if the total cost of subsidies becomes too high, the federal agent may lower subsidies, which affects all states.

**Hardness of reductions to and finding solutions of Stackelberg games.** Our setup with a federal agent adjusting the objectives of the state agents resembles a general-sum Stackelberg-like game, with the federal agent being a leader with $N$ followers. Here the leader solves a mechanism design problem. However, it is hard to derive exact solutions for these settings, even when attempting reductions for a single-step setting.

Reducing this problem to an *exact* single time-step problem is hard. The outcomes under the SIR dynamics are non-linear, vary in time, and depend in a non-trivial temporal way on the sequence of the leader and followers' actions. For instance, early lock-downs may dampen the pandemic, but lifting the lockdown too early may allow infections to spread again. Note also that the leader can act at each time-step in our simulation (with a 90-day freeze after it changes its decisions). Hence, a hypothetical one-time-step equivalent for our problem would at the very least equate each contingency of leader and follower actions to a single meta-action for the leader and the followers. That is, naively $|A_{\text{leader}}|^T$ leader actions and $|A_{\text{follower}}|^T$ follower actions, whose outcomes may vary in a highly non-obvious way. Second, such a reduction would need to approximate the effects of the sequential interplay between the leader and followers in our sequential game, but this is non-trivial to summarize.

Even if a suitable single time-step reduction was found, it remains computationally hard to solve general-sum Stackelberg games. Certain instances may be polynomial-time solvable with relatively stringent assumptions (Korzhyk et al., 2010), e.g., one assumes that the follower always plays the best response to the leader's action. However, it remains challenging when the followers face sequential, nonlinear dynamics or when the action space is exponentially large (Blum et al., 2019).

### 4.6 Data limitations make causal inference infeasible.

In our experimental setup, the observational data was generated using the historical policy played by the federal and state agents. In this work, our RL agents deviate from those policies.

Hence, there is a possibility that an unobserved (causal) factor could change the simulation dynamics when we deviate from the historical policy that generated the data. The current data is unfortunately too limited to do (interventional) measurements to determine whether such factors exist. The parameters are fitted to this pandemic and there has been only one set of government policies that have been rolled out. For the stringency level, there is some variation between states (some states were not as stringent as others); for the federal subsidy policy, there is no other data or governments to compare to. Hence, a causal analysis is unfortunately beyond the reach given the current data.

Given sufficient data, one might perform causal inference and eliminate confounding hidden causal factors, analyze "natural experiments", among others, that may help elucidate the full structure of the (simulation) dynamics (Pearl, 2003; Yao et al., 2020). This might more completely describe real-life outcomes under policies that are different from the data-generating policies. However, in lieu of such analysis, the SIR model is a well-established model for modeling pandemics, and hence is the best base model given the available data.

## 5 Results

We compare simulated outcomes using real-world and RL policies, trained using the AI Economist framework with the $\hat{\alpha}_i$ that maximize social welfare under the real-world policies.

Throughout training and analysis, we initialize each simulation episode such that $t = 0$ corresponds to March 22, 2020. During training, episodes run for $T = 540$ timesteps; however, at the time of this analysis, real-world data were only available through the end of April 30, 2021 ($t = 404$), so we treat this date as the end of our analysis window.

### 5.1 Policy Model

We use log-linear policies:

$$\pi(\sigma_{i,t} = j | o_{i,t}) = \frac{1}{Z_{i,t}} \exp\left(\sum_k o_{ik,t} W_{kj} + b_{ij}\right). \tag{17}$$

Here, $\pi(\sigma_{i,t} = j | o_{i,t})$ denotes the (conditional) probability that state $i$ will select stringency level $\sigma_{i,t} = j$ at time $t$, given its observations $o_{i,t}$. We discretize the stringency level $\sigma$ into 10 levels, such that $j \in \{1, \ldots, 10\}$ indexes each of the possible stringency levels. The normalization $Z_{i,t}$ ensures that $\pi$ is a proper probability distribution for all $i$ and $t$: $\sum_j \pi(\sigma_{i,t} = j | o_{i,t}) = 1$. The terms $W$ and $b$ represent the learnable parameters that are optimized when training the RL policies. $W_{kj}$ is the weight matrix, with $j$ indexing stringency levels and $k$ indexing input features. $b_{ij}$ is the bias that state $i$ has towards stringency level $j$. The weight matrix $W$ is shared across states, whereas the bias terms $b$ are specific to each state. A similar model is used for the federal policy. This policy model choice emphasizes explainability and simplicity, which we explore in more detail hereafter.

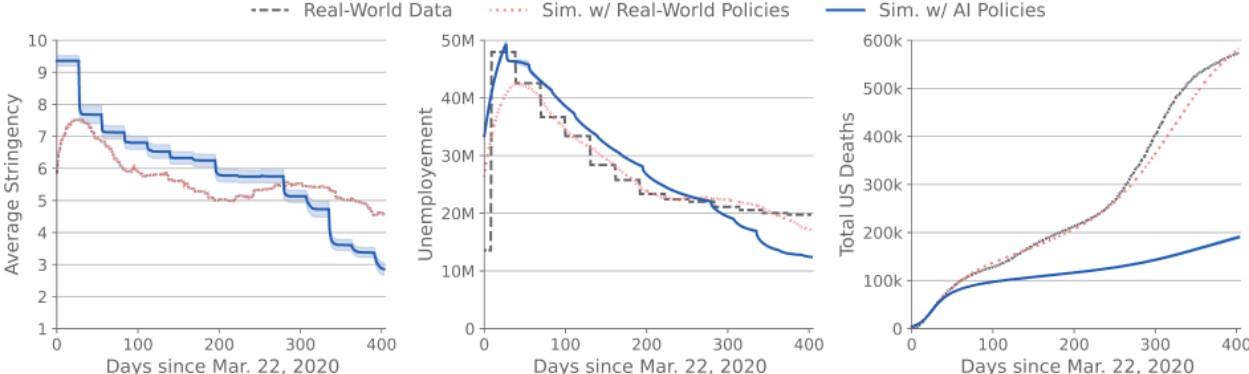

Figure 3: **Real-world vs simulation outcome.** Left: average state policy stringency level. Middle: US unemployment rate. Right: total US deaths. RL policies impose more stringent policy at the start of the pandemic before tapering off more quickly. This temporarily yields higher unemployment, but results in fewer total deaths. The simulation fits the data well when executed with the real-world policy.

## 5.2 Comparing to Simple Heuristic Policies

Simple heuristic baselines would be policies that always use 0% ("always open") or 100% stringency ("always closed"), or intermediate versions that increase and then decrease stringency after a certain amount of time. The outcomes of these simple policies are visualized in Figures 4 and 5. Generally speaking, compared to the real data, an "always open" policy would lead to higher deaths and infections, but also potentially higher economic index values. A "always closed" policy would have the reverse effect: lower deaths, lower infections, but also lower economic index values. However, they also yield lower social welfare compared to the real-world policies. As such, the real-world policies found a better health-economic trade-off (under our definition of the indices) and provide a strong baseline to compare the AI policies against.

## 5.3 AI vs Real-World Policies

Figure 3 compares RL and real-world policies and outcomes. Compared to real-world policies, RL policies (blue lines) impose comparatively higher stringency at the start of the outbreak but reduce stringency more rapidly. Similarly, RL policies result in more unemployment early on but recover towards pre-pandemic levels more quickly. Overall, however, unemployment under RL policies is higher on average during the analysis window. Moreover, RL policies result in considerably fewer COVID-19 deaths in this simulation. Figure 6 illustrates these trends for several states. For a full view including all states, see the Supplementary Material.

As a consistency check, simulated unemployment and COVID-19 deaths under real-world policies (dashed red lines) approximate the real-world trends well (dashed gray lines).

## 5.4 Improved Social Welfare

Figure 7 shows the percentage change in welfare, for each agent, from RL policies versus from real-world policies in the simulation. The top subplot shows results when social welfare is defined using the $\hat{\alpha}$ values obtained during calibration (see above).

Federal-level welfare is 4.7% higher under RL policies. We identify two features underlying this improvement. First, the AI stringency policies achieve a more favorable balance between unemployment and COVID-19 deaths. Second, the AI subsidy policy provides very little direct payments in order to achieve this balance.

In comparison, the real-world policies include a total of \$630B in subsidies (versus just \$35B on average for RL policies). State-level agents *gain* welfare via subsidies. As a result, the welfare benefits from RL policies

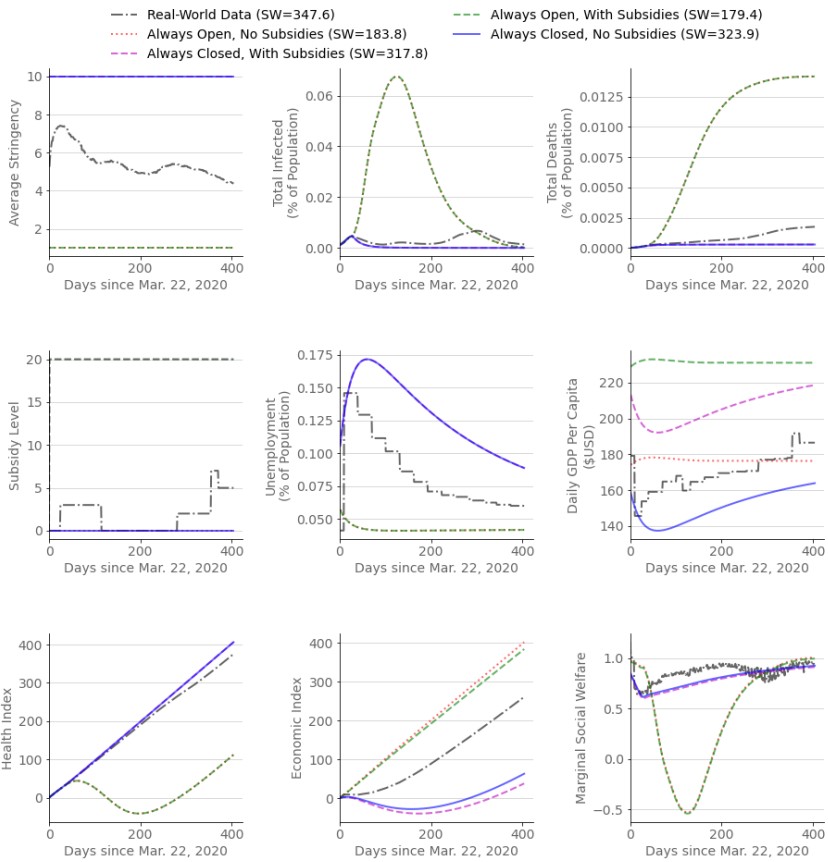

Figure 4: **Comparing heuristic always-open (no subsidies) and always-closed (with subsidies) baselines with real data.** Simple heuristic baselines include policies that always stay open (0% stringency) or always close (100% stringency), *applied uniformly across states*. For both cases, subsidies could be applied or not. A case that only focuses on maximizing the health index would always close and provide subsidies. On the other hand, if only focusing on the economy, one might always open and provide no subsidies. Here, we see that an always-open policy leads to significantly more infections (top row, middle) and deaths compared to always-closed policies, and hence lower health indices. Note that subsidies have no effect on the health index as the policies are not strategic and do not depend on the level of subsidies. Adding subsidies mechanically improves the economic index and GDP per capita. However, it does not change unemployment (which does not directly depend on subsidy levels). In comparison, the real-world policies used a mixture of stringency and subsidies, and yielded higher social welfare overall (347.6), therefore finding a better trade-off between health and economic indices (following our definitions). In contrast, always-closed and no subsidies was the best heuristic policy, achieving 323.9 social welfare.

are less pronounced at the level of states. Nevertheless, we observe that welfare improves for 33 of the 51 state-level agents when using RL policies.

Our framework also allows us to analyze outcomes under many possible definitions of social welfare, as we show below. To explore outcomes under different social welfare prioritizations, we set $\alpha_i$ to scale the relative prioritization $\frac{\alpha_i}{1-\alpha_i}$ by a factor $m_i$. For example, to explore the case where state $i$ cares twice as much

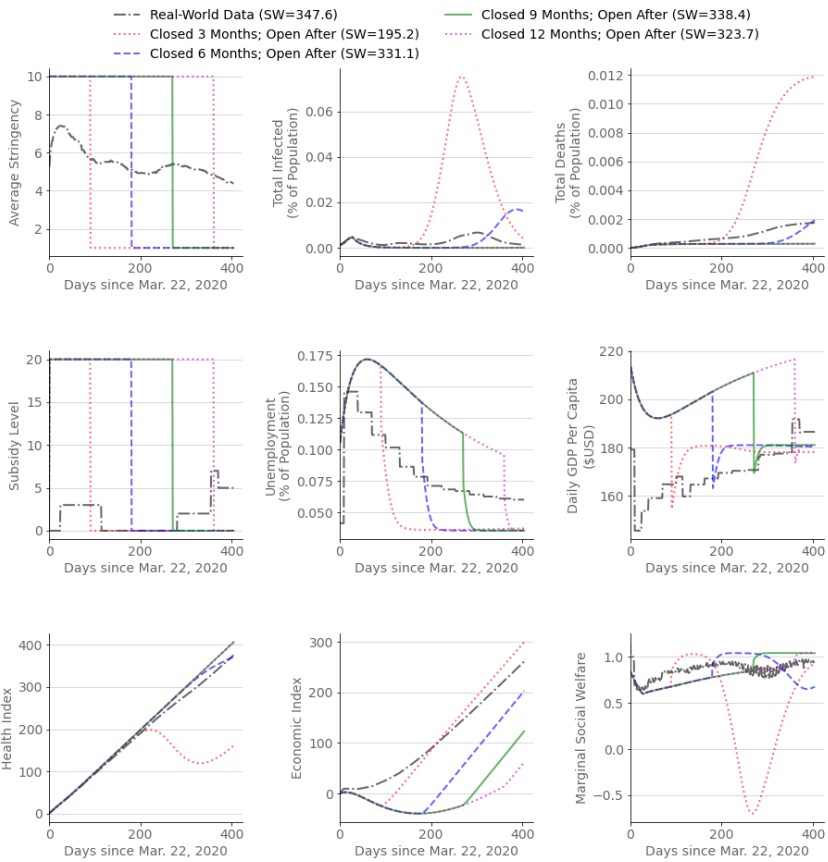

Figure 5: **Comparing closed-to-open policies baselines with real data.** Here, we compare variations of the always-open and always-closed policies as visualized in Figure 4. Specifically, we consider closing (100% stringency) for 3 to 12 months and then opening (0% stringency). Simultaneously, subsidies are maximal and then 0 in the same period. We see that lowering stringency after 3 months (red) leads to high infection and death rates later, compared to policies that lower stringency later. However, lowering stringency after 3 months does lead to higher economic indices, even compared to the real-world policy. Note that the social welfare of real-world policies (347.6) is higher than all the heuristic baselines (achieving 338.4 at most), analogous to the results in Figure 4. Hence, real-world policies found a better health-economic trade-off (using our definitions of indices) versus these baselines, and as such, suggest real-world policies are a strong baseline to compare our AI policies with (see Section 5.3 and Figure 6).

$(m_i = 2)$ about health outcomes compared to the real-world policy, we set $\alpha_i$ such that $\frac{\alpha_i}{1-\alpha_i} = 2 \cdot \frac{\hat{\alpha}_i}{1-\hat{\alpha}_i}$, where $\hat{\alpha}_i$ is the implied real-world value obtained from data.

Figure 7 (middle) shows welfare improvements using $m_p = 4$. In this case, the federal government's increased health prioritization leads to very large subsidy levels ($4.4T on average), which considerably improves state-level welfare. These large subsidies induce a shift towards higher stringency in the state-level policies and, hence, better health outcomes. In effect, the federal objective (when $m_p = 4$) benefits from this extreme subsidy level as well.

When the increased health priority is applied to the states ($m_{1:N} = 4$), we again see consistent social welfare improvements compared to real-world policies (Figure 7, bottom). In this case, outcomes improve due to

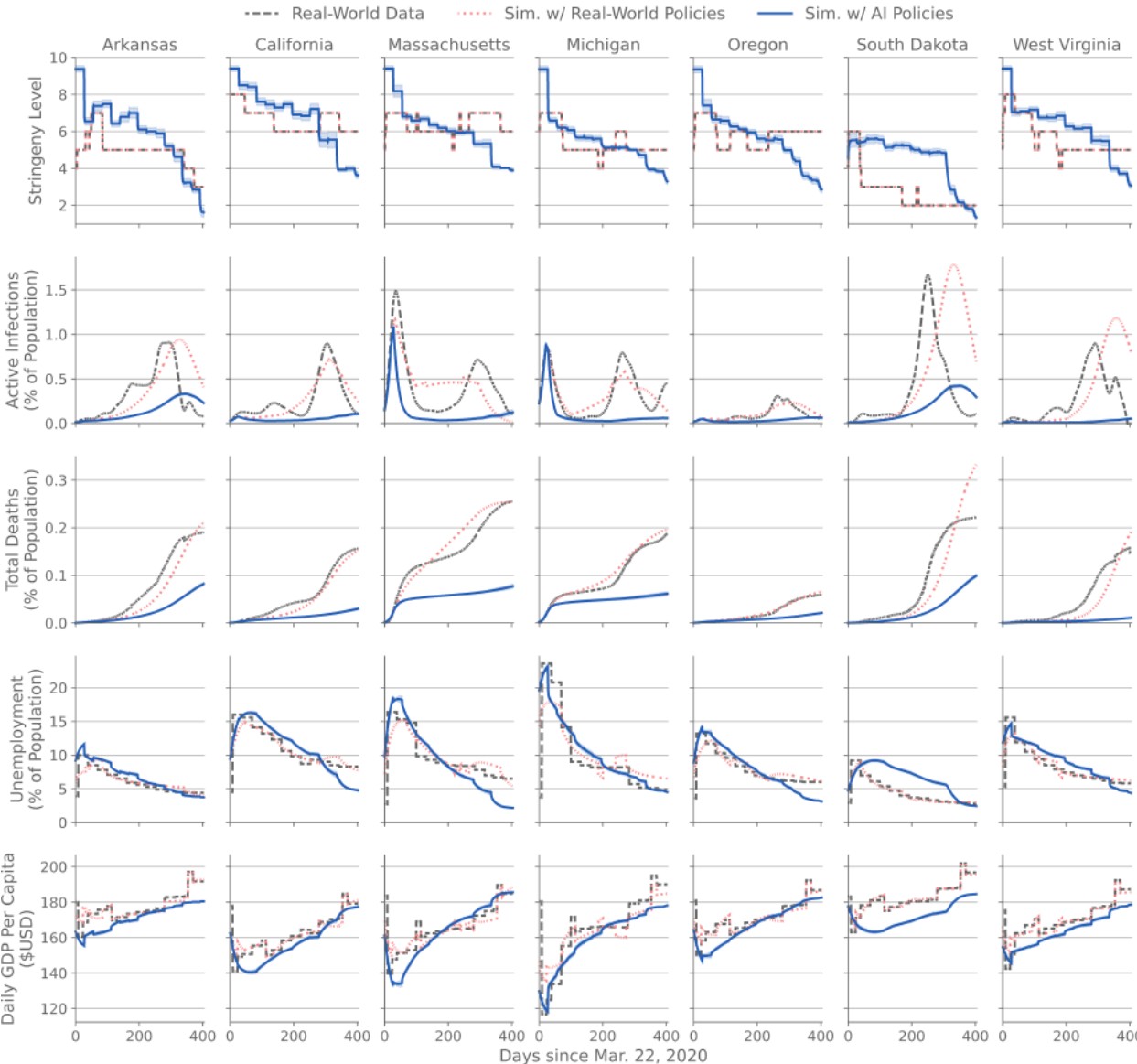

Figure 6: **Real-world vs simulation outcome at the state level.** In order, from top to bottom: Stringency level; active COVID-19 case load; cumulative COVID-19 deaths; unemployment; and daily economic output. Infections, deaths, and unemployment numbers are expressed as percentages of the state population; economic output is similarly normalized to reflect *per capita* daily GDP. Note: For the "Real-World" data (gray), infection numbers are estimated based on available data on COVID-19 deaths (see Methods) and economic output is estimated from our model given real-world data as inputs.

the capacity of the RL policies to adapt to different objectives, while the real-world stringency policies are (as expected) sub-optimal for this level of health prioritization.

### 5.5 Outcomes under Varying Welfare Objectives

Our framework can flexibly optimize policy for any quantifiable objective. This allows us to explore health and economic outcomes for a wide range of health prioritizations. To do so, we train RL policies across a range of welfare parameterizations $\alpha_i$ at the state-level and federal-level. We examine different re-scalings $m_i$ of the ratio $\frac{\hat{\alpha}_i}{1-\hat{\alpha}_i}$, where $m_i > 1$ captures that agent $i$ gives more weight to public health compared to

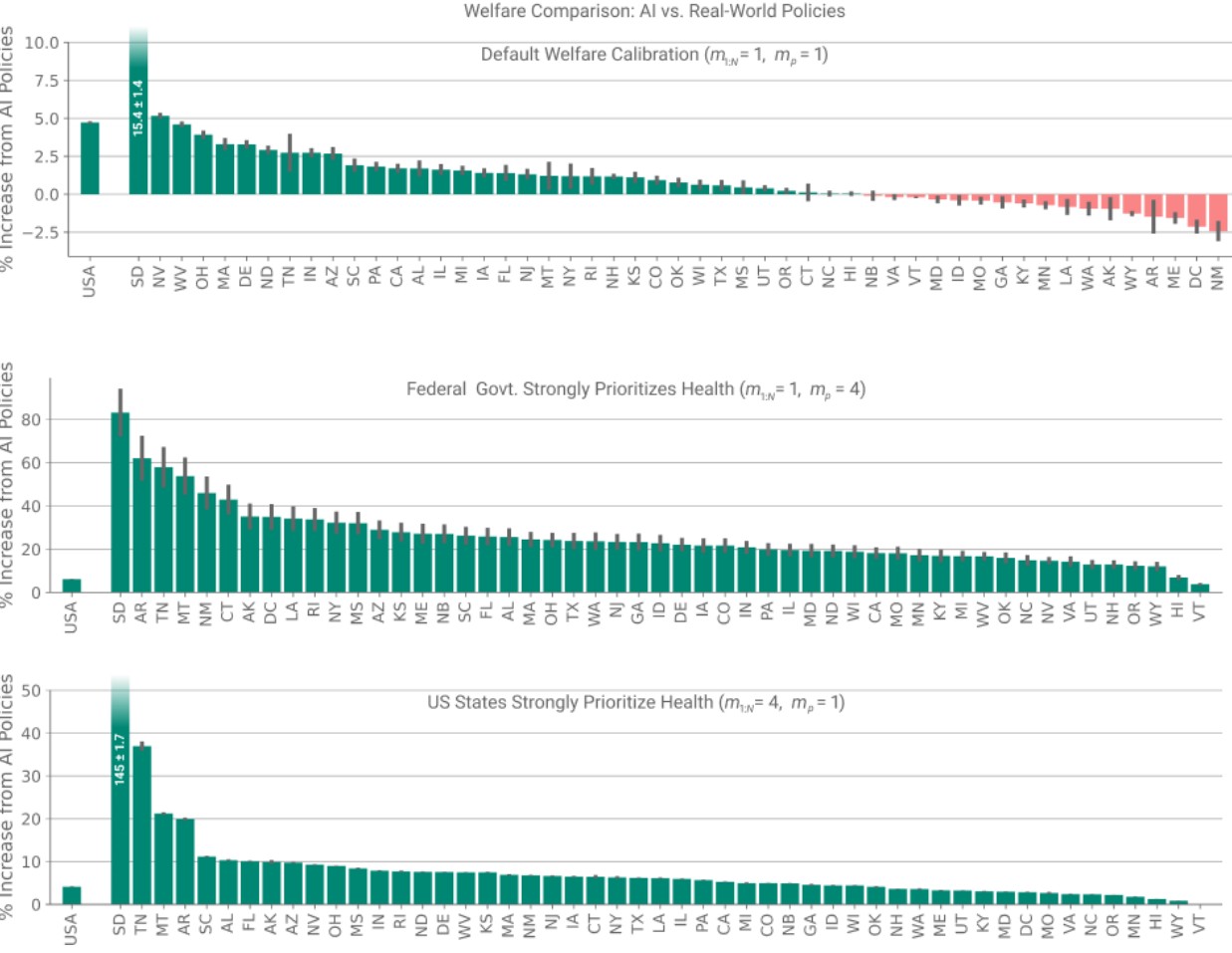

Figure 7: **RL policies Achieve Higher Social Welfare in Simulation.** RL policies achieve higher social welfare at the federal level and for 33 out of 51 states when compared to real-world policies. Top: welfare comparison when using $\hat{\alpha}$ values given by the default calibration. Middle: welfare comparison with highest tested $\alpha_p$ and default $\hat{\alpha}_{1:N}$. Bottom: welfare comparison with highest tested $\alpha_{1:N}$ and default $\hat{\alpha}_p$. In each plot, welfare is calculated with the $\alpha$ values used to train RL policies. Bar heights and error bars denote the mean welfare improvement (as a percentage of welfare under real-world policies) and STE, respectively, across the 10 random seeds used to train RL policies, for each agent in the simulation.

the default calibration $\hat{\alpha}_i$ (described above). Figure 8 shows the state-level (left) and federal-level (right) outcomes, in terms of health ($H_i$) and economic indices ($E_i$), under various settings for $m_{1:N}$ (relative health prioritization of each state) and $m_p$ (relative health prioritization of the federal government).

As expected, changing the policy objective leads the RL policies to select a different trade-off between public health and the economy. Higher $m_{1:N}$ lead to a higher health index at the expense of the economic index. A similar trend is seen at the federal level with increasing $m_p$. In our model, subsidies incentivize a state to be more stringent by reducing the economic burden of additional unemployment. For certain settings of $m_{1:N}$ and $m_p$, the federal government prefers to use this economically costly mechanism to better align states' incentives with its own policy objective. As a result, the trade-off between public health and the economy selected at the federal level depends on $m_p$. Interestingly, however, because subsidies reduce the trade-off states face between public health and the economy, higher $m_p$ tends to increase *both* Health and Economic Indices at the state level. However, this comes at a higher total borrowing cost to fund subsidies.

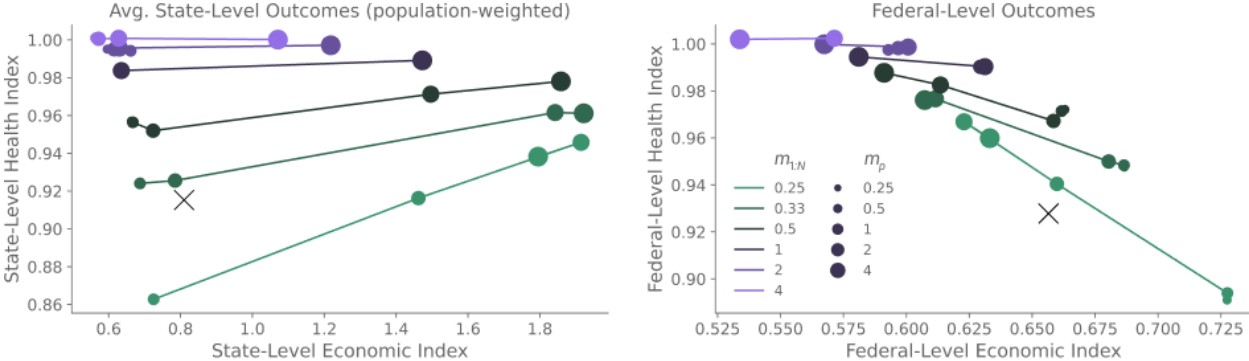

Figure 8: **Health and economic indices for different social welfare objectives.** Colors indicate the relative health priority scaling (defined in main text) $m_{1:N}$ of state policies. Dot sizes indicate the relative health priority scaling $m_p$ of federal policy. Black 'X' denotes index values achieved under the real-world policy. *Ceteris paribus*, as states emphasize health more (higher $m_{i:N}$, fixed $m_p$), state-level health indices increase on average, but the economic index decreases. Similarly, as federal policy emphasizes health more (higher $m_p$), state and federal health indices increase, but the federal economic index decreases, reflecting the higher borrowing cost of subsidies. At $m_{1:N} = 4$ (light purple), the federal health index stays constant as $m_p$ varies between 0.25 to 4 (size of the bullets). In this case, the states already overweight the health index in their optimization objective (compared to the inferred real-world balance $\alpha_i$). More subsidies may increase a state's economic index, but the effect of more subsidies in this case doesn't meaningfully improve a state's economic index objective even more, and states are already using the highest stringency level possible (and so can't do more to dampen the pandemic even faster).

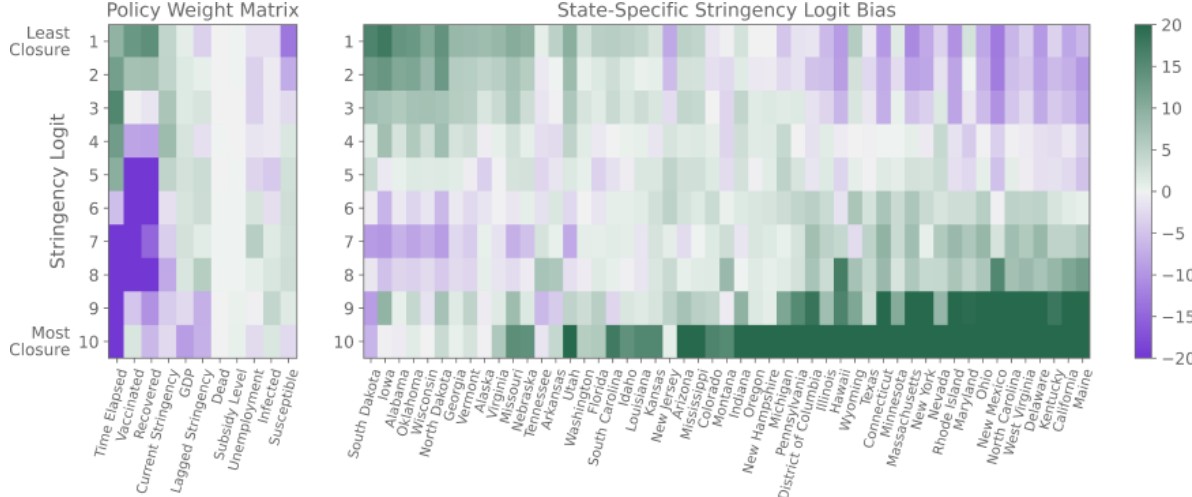

Figure 9: **Explaining policies using the learned weights of state-level RL policies.** Left: weights $W_{kj}$ for each input feature (columns indexed by $k$) and stringency level (rows indexed by $j$) show how learned state RL policies respond to pandemic conditions. For instance, as infections increase, stringency levels increase because $W_{kj} > 0$ for higher stringency levels and $W_{kj} < 0$ for lower stringency levels. Analogously, as recoveries increase, stringency levels decrease. Right: state-specific biases $b_{ij}$ show how states (columns indexed by $i$) have varying stringency level preferences.

## 6 Explaining Learned Policies

To build trust, (the behavior of) learned policies should be *explainable*. The simple structure of our log-linear policy enables *feature attribution* (Ribeiro et al., 2016), a form of *post-hoc explainability*. That is, the weights

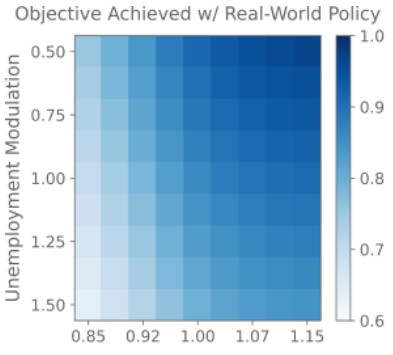 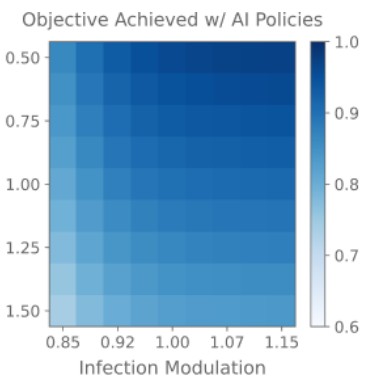 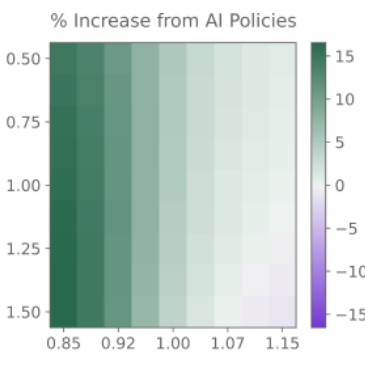

Figure 10: **Analyzing social welfare robustness under simulation parameter perturbations.** Left: federal-level social welfare $F_p$ under real-world policies, across a range of model perturbations affecting how unemployment (rows) and COVID-19 transmission rates (columns) respond to stringency policy. Perturbations modulate the scale of the response; for example, with unemployment modulation $> 1$, the same increase in stringency yields a larger increase in unemployment (see Methods Section 9.7 for details). Middle: same, but under RL policies. Right: $F_p$ under RL policies, as a percentage of $F_p$ under real-world policies. Social welfare improvements persist across a large range of perturbations.

of the policy show how its decisions depend on (a change in) the input features. Figure 9 visualizes the learned stringency policy weights $W_{kj}$ and $b_{ij}$ (Eq. 17), trained with health prioritizations $\hat{\alpha}_i$ and averaged over 10 repetitions with random seeds. Examining $W$ confirms several intuitions: 1) the learned policy is more likely to use higher stringency levels as susceptible and infected numbers increase; 2) analogously, they are more likely to use lower stringency levels as recovered and vaccinated numbers increase; and 3) in addition, the biases $b$ show how stringency level preferences vary across states.

Importantly, explainable policies enable identifying potentially *undesirable* input or policy features. For example, we can see that $W$ encodes a strong shift towards lower stringency as the time index grows higher ("time elapsed" in the Figure). However, timing indicators may be less semantically relevant than epidemiological or economic metrics. Hence, one may decide to regularize RL policies to rely less or not at all on the time index or other input features.

## 7 Robustness Analysis

Policies should be robust to calibration errors, noisy data, and other issues. To this end, we evaluate how well learned policies perform when perturbing the simulation parameters. Figure 10 shows the change in federal social welfare $F_p$ under perturbations of the unemployment rate and COVID transmission-rate response parameters. Intuitively, social welfare decreases for higher unemployment rate and lower transmission rate responses to changes in stringency level. This is true both for real-world policies (Fig. 10, left) and RL policies (Fig. 10, middle). However, social welfare is more robust to perturbations under RL policies. The gains from RL policies remain positive under most perturbations, except when unemployment and transmission-rate responses are strongly modulated (Fig. 10, right). The stringency and subsidy choices of RL policies differ in each perturbation setting because these perturbations affect the inputs to the policy networks.

We can perform similar analyses when varying other parameters, e.g., if the mortality rate doubles, subsidies would be higher and stringency levels would be higher. Similar results hold for lower recovery rates (infections linger longer and cause more infections per infected person).

This analysis shows that the learned RL policy continues to perform well across a significant range of perturbations.

# 8 Discussion

Our results show the AI Economist framework has strong potential for real-world policy design. Taken together, our results motivate building representative and fine-grained simulations and developing machine learning algorithms to learn AI-driven policies that can *recommend* strategies to improve social welfare objectives in the real world. With this goal in mind, we now discuss important considerations, limitations, and extensions of our framework.

## 8.1 Data

**Data Availability Constrains Simulation Model.** All modeling choices are informed by the availability of data or existing domain knowledge (or lack thereof). For example, we summarize pandemic response policy using a stringency level, as fine-grained details on the efficacy of individual policy levers is lacking. Hence, AI policy design can be improved with more and higher-quality data which would enable more fine-grained design choices, e.g., including more health, economic, and policy variables.

**Retrospective Analysis uses Future Data to Emulate Domain Knowledge.** We apply AI policies from the start of the COVID-19 pandemic, i.e., March 2020 onward, but we train AI policies in a simulator calibrated on data from all of 2020 *that was gathered under the real-world policy.* Hence, our analysis uses data that would not have been available in the real world, if hypothetically performed in March 2020. Our use of "future data" emulates the use of 1) scientific estimates of unknowns, 2) domain knowledge, and 3) previous experience. Together, such knowledge could provide forecasts of the pandemic and simulation parameter estimates, which in turn enable training AI policies.

**Coarse Level of Data Aggregation.** Policies may affect some social groups differently than others, but differential data on policy outcomes are not readily available for the COVID-19 pandemic. Hence, it is hard to accurately simulate the effects of real-world diversity with the current data. Our simulation models a version of the US where unemployment, the pandemic, and policy impact all people in the same way. Representing diversity is a necessary step if this technology is eventually used in the real world. However, this requires more robust fine-grained data than is readily available.

## 8.2 Methodology

**Definition of Social Welfare.** We defined social welfare as to capture the basic trade-off between public health and the economy. In general, social welfare can include any number of priorities which are outside the scope of this work. One could add, e.g., keeping ICU beds available, preventing businesses from failing, or minimizing inequality. Applying the AI Economist in the real world would require robust consideration of how social welfare is defined and input from a sufficient set of representative stakeholders.

**Convergence Guarantees.** The optimal RL policy satisfies the Bellman equation. Hence, one can find optimal policies using dynamic programming techniques. In specific circumstances, RL algorithms provably converge to this optimal policy, e.g., Q-learning in tabular settings with discrete states and actions. Reinforcement learning based on gradient descent implements a form of approximate dynamic programming: the theoretically optimal policy function for a given environment may not lie in the parametric function space, while gradient descent may not be guaranteed to converge to the optimal solution. This issue is exacerbated in the multi-agent setting. However, we show that our use of RL empirically finds policies that perform well, e.g., outperform strong baselines. It is an open challenge to provide general theoretical convergence guarantees for two-level, multi-agent RL.

## 8.3 Simulation Model

**Independency Assumptions across States.** Our simulation uses an independent SIR model for each region and does not model interactions and cross-over effects between states, i.e., it assumes that COVID-19

cases don't spread across state lines. This simplifies calibration, while already demonstrating a good fit with real-world data.

**Policy Impact Factors are Assumed Static.**  In our simulation, setting a particular stringency level will always lead to the same transmission rate. In reality, the effect of policies may depend on past policy choices and the duration of the pandemic in complex ways. For instance, it may be possible that after a long period of highly stringent policy and subsequent more relaxed policy, a second period of renewed stringent policy may not be as effective as public adherence may decrease due to fatigue. The lack of fine-grained data makes it difficult to model such subtleties. Adding such features could improve the realism of our simulation.

**Structural Estimation, Correlation, Causation, and Generalization.**  In our model, the unemployment model and infection rates depend on the stringency level of state policy only. This is an intuitive high-level modeling choice which yields strong out-of-sample performance given historical data. However, there is no (counterfactual) data to ensure this is not a spurious correlation, nor that including other causal factors may yield models that generalize better. In particular, our model extrapolates the behavior of unemployment and transmission rates (as a function of policy measures) to pandemic situations that are unseen in the real world. As such, real-world policy design in practice often entails continuous re-calibration and structural estimation.

**Modeling Human Behavior using Machine Learning.**  We assume that all agents behave rationally, i.e., optimize policy for a given definition of social welfare. However, in the real world agents may not behave optimally or their behavior may not be well explained by standard objective functions. For example, many human cognitive biases are known, such as recency bias, ownership bias, behavioral inattention (Gabaix, 2019), and are well studied in behavioral economics (Mullainathan & Thaler, 2000).

Hence, rational RL agents may not be sufficiently representative of the real world. Moreover, multi-agent learning algorithms may not always represent how real-world agents learn and adapt in the presence of others, or make unrealistic assumptions about how much agents know about the behavior of other agents. As such, it would be fruitful to explore extensions of our framework that include human-like learning and behaviors. However, it is an open challenge to collect sufficient micro-level data for these purposes.

### 8.4    Real-World Application

**Encoding Values and Objectives.**  Our work is descriptive, not normative. We show outcomes for a class of stylized social welfare objectives which follows standard economic modeling. However, real-world objectives may include many more factors, while it may not always be clear if and how those should be quantified.

**Explainability.**  As of yet, there is no general definition of explainability or interpretability (Doshi-Velez & Kim, 2017). Rather, explanations are often domain-specific and their perceived quality is subjective. Moreover, "plausible" explanations may not be robust (Slack et al., 2020) or generalize to all data or unseen environments when using high-dimensional models, such as deep neural networks.

**Ethics.**  This work should be regarded as a proof of concept. There are aspects of the real-world that no AI simulation can capture as of yet. As a result, the simulations proposed here, and the insights that come from them, are not designed to inform, evaluate, or develop real-world policy. All data have their limitations and those used to model complex systems like health and the economy may fail to model impacts on specific segments of the population (e.g. historically marginalized or other vulnerable groups). Our goal is to develop more realistic simulations in the future. For an extended ethics discussion, see the Supplementary Material.

## 9 Methods

### 9.1 Data and Model Calibration

Our simulation is grounded in real-world data. We combine publicly available data on state-by-state COVID-19 deaths, unemployment, and stringency of public health policy, as well as federal policy concerning direct payments.

**Policy Data.** We take the daily stringency estimates provided by the Oxford COVID-19 Government Policy Tracker (Hallas et al., 2021), discretized into 10 levels, as the real-world data for $\pi$. The date and amount of each set of direct payments issued through federal policy are taken from information provided by the COVID Money Tracker project (Committee for a Responsible Federal Budget).

**Public Health Data.** To fit the disease model, we use the daily cumulative COVID-19 death data provided by the COVID-19 Data Repository at Johns Hopkins University (Dong et al., 2020). We treat death data $D_{i,t}$ as ground-truth and solve for $S_{i,t}, I_{i,t}, R_{i,t}$, and $\beta_{i,t}$ algebraically using the SIR equations (Eqs. 2-7), given fixed estimates of the mortality rate $\mu = 0.02$ and recovery rate $\nu = \frac{1}{14}$. Estimating the missing data this way simply requires rearranging the SIR equations to express the unknown quantities in terms of "known" quantities such as $D_{i,t}, V_{i,t}, \mu$, and $\nu$. For example, $R_{i,t} = \frac{D_{i,t}}{\mu}$ and $I_{i,t} = \frac{R_{i,t+1} - R_{i,t}}{\nu}$, and so on.

The SIR estimates are useful for setting the starting conditions of the simulation; however, this inference procedure primarily serves to estimate each state's daily transmission rate $\beta_{i,t}$. This allows us to measure the relationship between (delayed) stringency $\pi_{i,t-d}$ and COVID-19 transmission. We set $d$ to 29 days, since the empirical correlation between stringency and transmission is strongest at this delay.

Finally, we use linear regression to fit the infection rate parameters of Equation 8 for each state, while regularizing the state-to-state variability in these parameters to help prevent individual states from overfitting to noise.

We set the onset date for vaccinations and their daily rate of delivery to approximately match aggregate vaccination trends in the US. Specifically, vaccines become available in the simulation after January 12th, 2021, at a rate of 3k new daily vaccines per 1M people.

**Economic Data.** Monthly unemployment rates for each state are collected from the Bureau of Labor Statistics (Bureau of Labor Statistics, 2021); in the daily representation of these data, each day in a given month uses the reported value for that month. We fit the unemployment parameters $\lambda_k$, $w_{i,k}$, and $U_i^0$ (for each $i$, $k$) by minimizing the squared error between predicted daily unemployment rates and those contained in the data. As with the fit to transmission rate, we regularize the variability in $w_{i,k}$ across states.

**Train-Test Splits.** The simulation is calibrated on data from 2020. We use data through November 2020 to estimate model parameters and the remaining 2020 data to tune hyperparameters. In particular, we confirmed that, when applying the real-world stringency policies, the calibrated simulation predicts deaths and unemployment level that are consistent with real-world outcomes throughout 2020 and through April 2021.

Within the 2020 calibration data, predicted outcomes capture 80% and 39% of the population-normalized variance in unemployment and COVID-19 deaths, respectively. For the 2021 "test" data, the predictions capture 54% of unemployment variance and 35% death variance.

**Using future data to fit the dynamics and train RL agents.** Note that the SIR dynamics are fitted to data (from 2020), while the RL agents are trained on that same time-frame. However, in a real-world deployment, during 2020 a hypothetical RL agent would not have access to all the future data. However, in our experiment, the use of future data for training is a proxy for using experts' forecasts, domain knowledge, or experience from previous pandemics. That is, in real life, an RL agent could be trained on simulations whose parameters would be partially based (or estimated) using those auxiliary methods.

## 9.2 Social Metrics and Indices

Grounding the simulation in real-world data reveals a tradeoff between minimizing the spread of COVID-19 and minimizing economic hardship, such as unemployment. To quantify health and economic outcomes under various policy choices, we define a Health Index $H$ and Economic Index $E$. Our framework does not place any constraints on how social welfare $F_i$ is defined, but, for the purposes of this study, we define it to be a weighted sum of the Health Index and the Economic Index:

$$F_i = \alpha_i H_i + (1 - \alpha_i) E_i, \tag{18}$$

where $0 \leq \alpha_i \leq 1$ is a mixing parameter that captures the relative prioritization of health outcomes over economic outcomes for agent $i$.

The episode-averaged index values are the average of *normalized marginal index values* throughout the simulation episode:

$$H_i = \frac{1}{T} \sum_{t=1}^{T} \Delta H_{i,t}, \qquad E_i = \frac{1}{T} \sum_{t=1}^{T} \Delta E_{i,t}. \tag{19}$$

Intuitively, the Health Index decreases with the number of deaths and the Economic Index increases with overall economic output. We denote the unnormalized, marginal Health and Economic Index of state $i$ at time $t$ as $\Delta \tilde{H}_{i,t}$ and $\Delta \tilde{E}_{i,t}$, respectively. They are defined as:

$$\Delta \tilde{H}_{i,t} = -\Delta D_{i,t}, \qquad \Delta \tilde{E}_{i,t} = \mathrm{crra}\left(\frac{P_{i,t}}{P_i^0}\right). \tag{20}$$

The baseline productivity $P_i^0$ is the daily economic output expected under baseline, i.e., non-pandemic, conditions. We include a CRRA nonlinearity (Arrow, 1971) in the Economic Index, where $\mathrm{crra}(x) = 1 + \frac{x^{1-\eta}-1}{1-\eta}$ and $\eta = 2$ is a shape parameter. As a result, there are diminishing marginal returns on economic output.[6]

Indices for the federal government are defined similarly, but sum over the entire country and reflect the borrowing cost of any direct payments:

$$\Delta \tilde{H}_{p,t} = -\sum_{i=1}^{N} \Delta D_{i,t}, \tag{21}$$

$$\Delta \tilde{E}_{p,t} = \mathrm{crra}\left(\frac{\sum_{i=1}^{N} P_{i,t} - c \cdot \tilde{T}_{i,t}^{state}}{\sum_{i=1}^{N} P_i^0}\right), \tag{22}$$

where $c \geq 1$ is the borrowing cost of providing \$1 of direct payments. Note that $P_{i,t}$ already includes money received through direct payments $\tilde{T}_{i,t}^{state}$, so additional payments will always increase the Economic Index at the state level and decrease it at the federal level.

To standardize analysis, we normalize the marginal indices based on their minimum and maximum values, giving normalized marginal indices $\Delta H_{i,t}$ and $\Delta E_{i,t}$ for each agent $i$, as:

$$\Delta H_{i,t} = \frac{\Delta \tilde{H}_{i,t} - \Delta H_i^{\min}}{\Delta H_i^{\max} - \Delta H_i^{\min}}, \quad \Delta E_{i,t} = \frac{\Delta \tilde{E}_{i,t} - \Delta E_i^{\min}}{\Delta E_i^{\max} - \Delta E_i^{\min}}. \tag{23}$$

We obtain these minimum and maximum values from the average marginal indices measured by running the simulation under 2 policy extremes: minimum-stringency (i.e., fully-open) and maximum-stringency (i.e., fully-closed). The minimum-stringency policy contributes the minimum and maximum marginal Health and Economic indices, respectively, and vice versa for the maximum-stringency policy. When normalizing this

---

[6]The motivations for this modeling choice are discussed in the main text.

way, each minimum-stringency policy yields $H_i = 0$ and $E_i = 1$ and each maximum-stringency policy yields $H_i = 1$ and $E_i = 0$. As a result, higher index values denote more preferable outcomes.

Each state and federal agent aims to maximize the social welfare $F_i$ of its jurisdiction. Therefore, when applying RL, the instantaneous reward function is therefore the weighted sum of the normalized marginal indices:

$$r_{i,t} = \alpha_i \Delta H_{i,t} + (1 - \alpha_i) \Delta E_{i,t}. \tag{24}$$

### 9.3 Calibrating Policy Priorities from Data

To facilitate comparison between real-world policies and policies learned in our framework, we optimize AI policies for the social welfare with health priority parameters $\alpha_i$ that best fit real-world policies. Of course, it is very hard to know the exact mathematical form of the objectives of real-world agents. Rather, we attempt to identify the parameters $\hat{\alpha}, \hat{\alpha}_i$ that best explain the outcomes achieved by the real-world policy, assuming they used the social welfare objective as defined in Equation 18.

To obtain this estimate, we first collect simulated health/economic outcomes under 3 policies: the actual stringency choices, minimum stringency, and maximum stringency. We use these outcomes to estimate the Pareto frontier in the $(H_i, E_i)$ coordinate space. By definition, the $(H_i, E_i)$ coordinates for the minimum- and maximum-stringency policies define the endpoints of this frontier, at $(0, 1)$ and $(1, 0)$, respectively. We assume the Pareto frontier for state $i$ has form $E_i = (1 - H_i)^{x_i}$ and that the coordinates associated with the actual-stringency policy are found along this frontier. We set the shape parameter $x_i$ based on this latter assumption, and take $\hat{\alpha}_i$ as the value that maximizes social welfare along the estimated Pareto frontier:

$$\begin{aligned} \hat{\alpha} &= \max_{\alpha_i} \alpha_i H_i(\boldsymbol{\pi}) + (1 - \alpha_i) E_i(\boldsymbol{\pi}, H_i) \\ &= \max_{\alpha_i} \alpha_i H_i(\boldsymbol{\pi}) + (1 - \alpha_i) (1 - H_i(\boldsymbol{\pi}))^{x_i}, \end{aligned} \tag{25}$$

where the Economic and Health Index values are obtained from running the policies $\boldsymbol{\pi}$ in the simulation. In other words, given the estimate of the Pareto frontier, we find the $\hat{\alpha}_i$ that best rationalizes the outcomes obtained under the actual policy, i.e. the $\hat{\alpha}_i$ under which these outcomes are considered optimal.

When obtaining these estimates, we simulate the same period of time used for calibrating the simulation. As such, we calibrate $\hat{\alpha}_i$ from outcomes measured after running the simulation from March 23, 2020 to December 31, 2020.

Assuming a functional form for the social welfare objective and inferring $\hat{\alpha}$ for real-world policies facilitates comparing AI and real-world policies, and is not a complete view of what social welfare consistitutes in the real world. We emphasize that future use of AI policy design frameworks would involve first specifying policy priorities, e.g., what social welfare means, as opposed to inferring them *post-hoc*.

### 9.4 Multi-Agent Reinforcement Learning

We learn the optimal policy for all agents using multi-agent RL (Sutton & Barto, 2018). We use the term *actor* to refer to the two levels of agents: state governments (agents) and the federal government (social planner). Agents learn by interacting with a simulation environment, which iterates between states using dynamics $\mathcal{T}(\boldsymbol{s}_{t+1}|\boldsymbol{s}_t, \boldsymbol{a}_t)$, where $\boldsymbol{s}_t$ and $\boldsymbol{a}_t$ denote the collective states and actions of the agents at time step $t$. For each time step $t$, the social planner receives an observation $o_{p,t}$, executes an action $a_{p,t}$ and receives a reward $r_{p,t}$. Similarly, each agent $i = 1, \ldots, N$ receives an observation $o_{i,t}$, executes an action $a_{i,t}$ and receives a reward $r_{i,t}$. Note that each agent does not instantly observe action $a_{p,t}$ of the planner in its observation $o_{i,t}$. However, agents $i$ may see the effect of the planner's action $a_{p,t}$ at later times $t' > t$, e.g., if the planner increased subsidies for the next 90 days. Once all agents have acted, the environment transitions to the next state $s_{t+1}$, according to the transition distribution $\mathcal{T}$.

**Policy Models.** Each agent learns a policy $\pi$ that maximizes its $\gamma$-discounted expected return. We denote state policies as $\pi_i(a_{i,t}|o_{i,t}; \theta_i)$ and federal policy as $\pi_p(a_{p,t}|o_{p,t}; \theta_p)$. Here, $\theta_i$ and $\theta_p$ parameterize the poli-

cies. Let $\boldsymbol{\pi} = (\pi_1, \ldots, \pi_N, \pi_p)$ denote the collection of all policies and $\boldsymbol{\pi}_{-j} = (\pi_1, \ldots, \pi_{j-1}, \pi_{j+1}, \ldots, \pi_N, \pi_p)$ denote the joint policy without agent $j$.

Through RL, agent $j$ seeks a policy to maximize its expected reward:

$$\max_{\theta_j} \mathbb{E}_{a_j \sim \pi_j, \boldsymbol{a}_{-j} \sim \boldsymbol{\pi}_{-j}, s' \sim \mathcal{T}} \left[ \sum_{t=0}^{T} \gamma^t r_{j,t} \right], \tag{26}$$

with discount factor $\gamma \in (0, 1)$. The expected reward depends on the behavioral policies $\boldsymbol{\pi}_{-j}$ of the other agents and the environment transition dynamics $\mathcal{T}$. As such, Equation 26 yields an agent policy that best respond to the policies of other agents, given the dynamics of the simulated environment and the agent's observations.

**Sharing Weights.** For data efficiency, all $N$ state agents share the same parameters during training, denoted $\theta$, but condition their policy $\pi_i(a_i|o_i; \theta)$ on agent-specific observations $o_i$, which includes their identity. In effect, if one agent learns a useful new behavior for some part of the state space then this becomes available to another agent. At the same time, agent behaviors remain heterogeneous because they have different observations.

**Agent States and Actions.** Each agent $i$ (representing the governor of state $i$) chooses a stringency level $\pi_i \in [1, \ldots, 10]$, where $\pi_i = 1$ and $\pi_i = 10$ denote the minimum and maximum stringency levels, respectively. The agent's observation is:

$$o_{i,t} = (i, S_{i,t}, I_{i,t}, R_{i,t}, D_{i,t}, V_{i,t}, U_{i,t}, P_{i,t}, \pi_{i,t}, \pi_{i,t-d}, \tilde{T}_{i,t}^{state}), \tag{27}$$

which includes a one-hot encoding of the agent's index, the states of the SIR components, unemployment, (post-subsidy) productivity, current and delayed stringency, and the current level of subsidy provided by the federal government (see below).

**Planner States and Actions.** The planner chooses a subsidy level $\pi_p \in [1, \ldots, 20]$ that controls the amount of *direct payments* provided to workers. Each timestep, some amount of money is added directly to each state's daily productivity $P_{i,t}$, and that amount depends on the subsidy level and the state's population. At the minimum subsidy level, no money is added. At the maximum, each state receives a daily subsidy of roughly \$55 per person, corresponding to a direct payment rate of \$20k per person per year. The planner observes:

$$o_{p,t} = (S_t, I_t, R_t, D_t, V_t, U_t, P_t, \pi_t, \pi_{t-d}, \tilde{T}_t^{state}). \tag{28}$$

The planner observes the same types of information as the agents, but for all states at once. For instance, $S_t$ denotes that the planner observes the 51-dimensional array of susceptible rates for each state.

The planner's action space was designed to reflect a bandwidth of payments that includes the existing stimulus payments and to allow the federal RL agent to go beyond those amounts (to a reasonable degree). In the real world, subsidies were on the order-of-magnitude of USD 3000 per household (Department of the Treasury) in a single payment. Up to 2021, there had been 2 subsidy payments. Assuming a check is used for about a year, a real-world stimulus check could amount to about USD 3000/365 USD 8.22/day/household. An average US household has 2.5 people, hence the payment would be about USD 3.28/day/person. So daily subsidies of USD 0-55 correspond to roughly an order-of-magnitude larger checks potentially written by the RL agent, while 20 increments mean that the granularity is about USD 55/20 = USD 2.75 per subsidy level, corresponding roughly to 1.5 historical payment check. Of course, this depends on the assumption over what timespan a check is used on average (could be much shorter than 1 year), and multiple checks could be (and have been) issued in the real world. Hence, this range is reasonable to test whether RL agents would go far beyond (or below) what happened in the real world.

### 9.5 Constraining Policies for Improved Realism

To improve realism and the real-world viability of learned policies, we constrain RL policies to change slowly, as follows:

- We restrict the frequency with which each AI governor can update its stringency $\pi_i$. Specifically, if agent $i$ acts so as to change the stringency on timestep $t$, it is prevented from making any further changes until timestep $t + 28$. In other words, the stringency level for a state can change at most once every 28 days.

- The planner updates the level of direct payments $\pi_p$ every 90 days.

Future research could explore other constraints on policies, e.g., only allowing one increase in stringency level, and subsequently only decreases.

### 9.6 Two-level RL Strategies

Following (Zheng et al., 2021), we use *entropy regularization* (Williams & Peng, 1991) to stabilize simultaneous agent and planner optimization. Specifically, we schedule the amount of entropy regularization applied to the planner policy such that the planner policy is essentially random during the early portion of training. As a result, agents learn to conditioned policies on a wide range of possible subsidies while the planner's entropy regularization coefficient is gradually annealed to its final value. After this annealing, both the agents and the planner are able to stably optimize their policies towards their respective objectives.

### 9.7 Sensitivity Analysis

We perform a sensitivity analysis to examine how systematic errors in model calibration affect simulated outcomes. In particular, this is useful to examine the possible range of calibration errors within which AI policies outperform real-world policies. We examine sensitivity to systematic under-/over-estimations of two factors: how much (1) the COVID-19 transmission rate and (2) the excess unemployment respond to the stringency policy. To simulate these conditions we replace $\beta_i^\pi$ with $m_\beta \cdot \beta_i^\pi$ and replace $w_{i,k}$ with $m_w \cdot w_{i,k}$, where $m_\beta$ and $m_w$ denote the amount of *modulation* of the transmission rate response and unemployment response, respectively. For example, $m_w = 1.5$ emulates the condition that the *actual* unemployment response is 50% larger than in the default calibration.[7]

We set $m_\beta = 1$ and $m_w = 1$ during training. When performing the sensitivity analysis, we collect simulated outcomes across a grid of $0.85 \leq m_\beta \leq 1.15$ and $0.5 \leq m_w \leq 1.5$ values, for both the real-world policies and with the AI policies. Note that the real-world policies are fixed, but the AI policies may change for different $m_\beta$ and $m_w$ since these impact the observations on which the AI policy decisions are based.

### Broader Impact Statement

Policy making is a critical tool to address complex socioeconomic challenges, such as responding to pandemics. However, policy design is a significant technical challenge, because 1) policy making is often decentralized across agents whose incentives may differ; 2) agents who are affected by policy may change their behavior; and 3) analytical and computational methods quickly become intractable or need unrealistic stylized models. Our results show that reinforcement learning offers a holistic solution to address these issues. Log-linear policies trained using RL achieve better health and economic outcomes than real-world policies in a grounded COVID-19 simulation. These policies are also simple (and thus implementable), explainable, and robust. As such, AI is a promising tool to guide real-world policy making.

This work should be regarded as a proof of concept. There are aspects of the real-world that no AI simulation can capture as of yet. As a result, the simulations proposed here, and the insights that come from them, are not designed to inform, evaluate, or develop real-world policy. All data have their limitations and those used to model complex systems like health and the economy may fail to model impacts on specific segments of the population (e.g. historically marginalized or other vulnerable groups). Our goal is to develop more realistic simulations in the future.

We are very clear that there are known and unknown risks in the publication of research around the use of AI to weigh the policy tradeoffs in potentially sensitive areas like health, employment, education and

---

[7]For instance, if actual unemployment numbers are under-reported in the calibration data.

the environment. In recognition of the potential risks, we commissioned Business for Social Responsibility (BSR) to conduct an ethical and human rights impact assessment of this research before its release.

Some, though not all, of the risks they noted are associated with the use of the simulation beyond its intended purpose and over-reliance on, or overconfidence in, its AI-driven policy recommendations. In BSR's words, they are as follows:

Policymakers may use the simulation and/or underlying research to inform decision-making. There is a risk that policymakers may use the simulation directly to test potential policies. Policymakers may also use insights and findings from published reports or press coverage to inform public policy.

The simulation and research may be used for purposes other than those originally intended. There is a risk that the limitations of the simulation and guidance on appropriate use are misunderstood, misinterpreted, or ignored by readers, leading to inappropriate usage of the simulation for purposes other than intended. This could include using the simulation to create policy responses for future cases not covered by the simulations conducted here.

Humans may over-rely on insights from the simulation. A simulation provides a veneer of precision and objectivity that imbues a high level of confidence in AI policies. This may result in humans using outputs directly, without external validation or review, and without testing the recommendations in real-world scenarios. Humans often over-rely on AI, sometimes with grave results. This is particularly concerning in the context of a "moral" or "ethical" tradeoff between economic health and public health and safety.

The simulation may be manipulated to 'optimize' for specific policies or outcomes. Policymakers or government actors may use a simulation to justify policy decisions, giving people a false sense of confidence in the policy without insight into the limitations of the simulation. Alternatively, policymakers may use open-source code to develop new models that optimize for their own self-interest, tweaking parameters and outcomes until they achieve the desired outcome.

The AI Economist open-source code may be altered for use in new geographies or to guide policy responses in future pandemics. If the underlying code and datasets for the AI Economist's COVID simulation are made publicly available, there is a risk that it could be altered for use in other geographies without the appropriate data inputs, or that it could be used to guide policy making and decisions in future pandemics, without the appropriate data on the disease.

These risks could be associated with the following adverse human rights impacts:

**Right to Equality and Non-Discrimination (UDHR Article 2 / ICCPR Article 2):** Failure to disaggregate data by race, gender, age, or other protected categories, limits a simulation's ability to extrapolate policy impacts on specific groups or populations. If a simulation is used to make policy decisions without taking into consideration policy impacts on specific segments of the population, there is a risk of discrimination and adverse human rights impacts on higher risk populations.

**Right to Life and Right to Health (UDHR Articles 3, 25 / ICCPR Article 6 / ICESCR Article 12):** If a policymaker or government actor uses outcomes from a simulation to justify less stringent public health policies, it could lead to severe adverse health impacts, including long term illness or death. Furthermore, if the underlying code is used to create simulations that inform policy decisions in future pandemics, it may lead to similar adverse health impacts, particularly if the new model fails to integrate data sources and context specific to the disease.

**Right to Work and Adequate Standard of Living (UDHR Articles 23, 35 / ICESCR Article 6):** Policymakers and government actors may use the model in ways that result in job loss and adverse impacts on the right to work. For example, if a policymaker or government actor uses the simulation to justify stringent public health measures that lead to lockdowns, this could result in significant job loss. These impacts may be particularly severe if the policy response does not include financial support or subsidies for individuals experiencing financial hardship. Violations of these rights may also have knock-on impacts on other human rights.

**Freedom of Movement, Freedom of Assembly, and Access to culture (UDHR Articles 13, 20, 27 / ICCPR Articles 12, 21, 27):** In addition to the rights above, stringent public health policies, including lockdowns, will impact individuals' freedom of movement, freedom of assembly, and access to cultural life of the community.

**Right to Education, Healthcare, and other Public Services (UDHR Articles 26, 25, 21b / ICESCR Articles 13, 15):** Depending on the context, public health policies may also impact access to education, healthcare, or other public services.

These adverse impacts are more likely to affect at-risk and vulnerable populations with less access to resources that would allow them to sustain themselves through job loss, economic shutdowns, and health complications.

To mitigate these risks to the extent possible, we have taken the following steps:

**Limiting the public web demo to insights.** The public website does not include any interactive elements, which allows individuals to optimize for variables or tradeoffs, so as to limit its potential use as a policy-making tool. Instead, it includes a summary of research findings and accompanying visuals.

**Gating the code.** Before gaining access to the simulation code, we have required an individual to provide their name, email address, affiliation and intended use of the code. In addition, we are asking users to attest to a Code of Conduct in its use. By doing so, we hope to significantly mitigate risks of misuse, abuse, or use beyond originally intended purposes, while still enabling the sharing of research to advance the field and providing the opportunity for replicability.

**Detailing limitations.** We have made every effort to be as transparent and clear as possible about the limitations of this simulation and the datasets which inform it. Our disclaimers include the intended use of the simulation, its limitations in modeling real-world effects, and the lack of disaggregated datasets.

**Creating a simulation card.** We have published a simulation card that provides a description of the COVID simulation's data sources and inputs, limitations, and presents basic performance metrics. It also provides 1) a concise overview of the simulation and accompanying research, its key audience, and intended use cases, 2) a description of the limitations and assumptions (both explicit and implicit) of the model, 3) clear definitions of the variables and descriptions of what the variables mean, as well as 4) clear communications on how it works. We hope that "model cards" and "simulation cards" of this nature become a standardized practice across technology companies and research institutions.

## Author Contributions

## Acknowledgments

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

# A   Appendix

| Parameter | Symbol | Value |
|---|---|---|
| Population total for state $i$ | $n_i$ | |
| The number of people in state $i$ that are **susceptible** at time $t$ | $S_{i,t}$ | |
| The number of people in state $i$ that are **infected** at time $t$ | $I_{i,t}$ | |
| The number of people in state $i$ that are **recovered** at time $t$ | $R_{i,t}$ | |
| The number of people in state $i$ that are **vaccinated** at time $t$ | $V_{i,t}$ | |
| The number of people in state $i$ that have **passed away** at time $t$ | $D_{i,t}$ | |
| Infection rate | $\beta_{i,t}$ | |
| Recovery rate | $\nu$ | $\frac{1}{14}$ |
| Mortality rate (the fraction of 'recovered' people that pass away) | $\mu$ | 2% |
| Stringency index | $\pi_{i,t}$ | |
| Productivity | $P_{i,t}$ | |
| State subsidy | $\tilde{T}_{i,t}^{state}$ | |
| Unemployed | $U_{i,t}$ | |
| Working age rate | $\nu$ | 60% |
| Daily productivity per worker | $\kappa$ | \$320.81 |
| Fraction of **infected** individuals that cannot work | $\eta$ | 0.1 |
| Health index | $H_i$ | |
| Economic index | $E_i$ | |
| Health welfare weight | $\alpha_i$ | |
| Social welfare | $F_i$ | $\alpha_i H_i + (1 - H_i)E_i$ |

Table 1: Model variables and parameters.

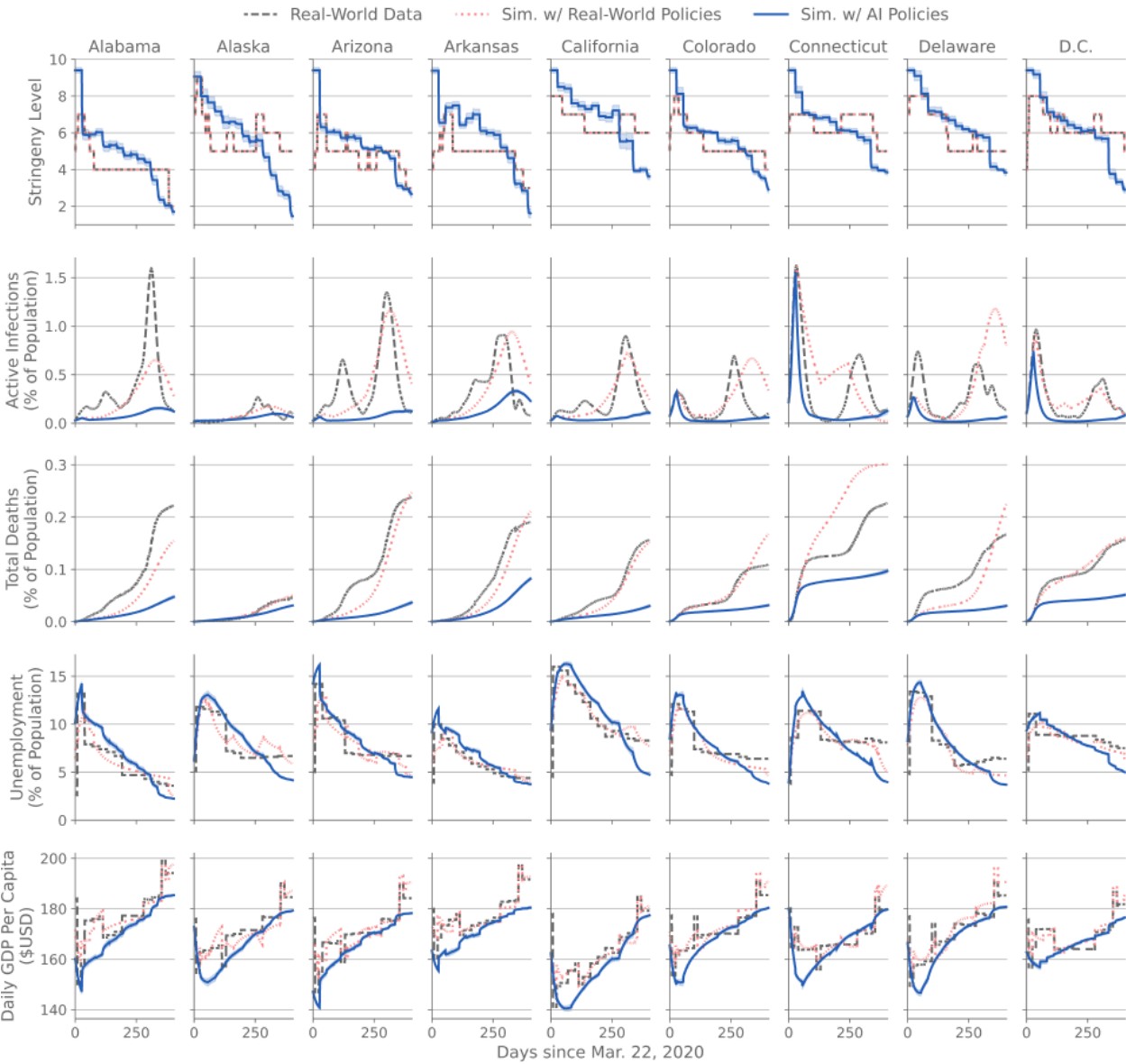

Figure 11: **Real-world vs simulation outcome at the state level (Alabama - D.C.).** Conventions are the same as Figure 6 in Section 5 of the main text.

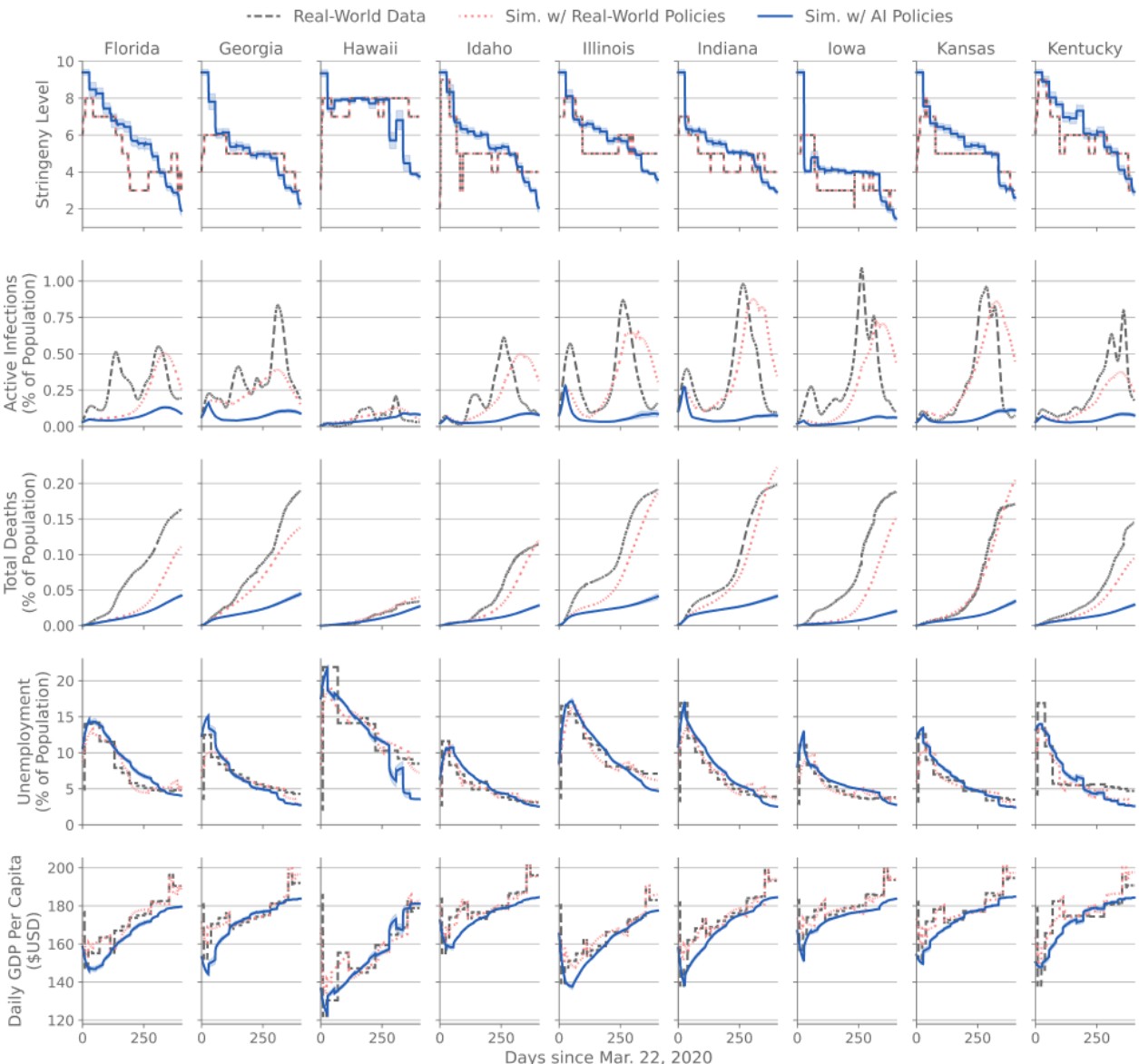

Figure 12: **Real-world vs simulation outcome at the state level (Florida - Kentucky).** Conventions are the same as Figure 6 in Section 5 of the main text.

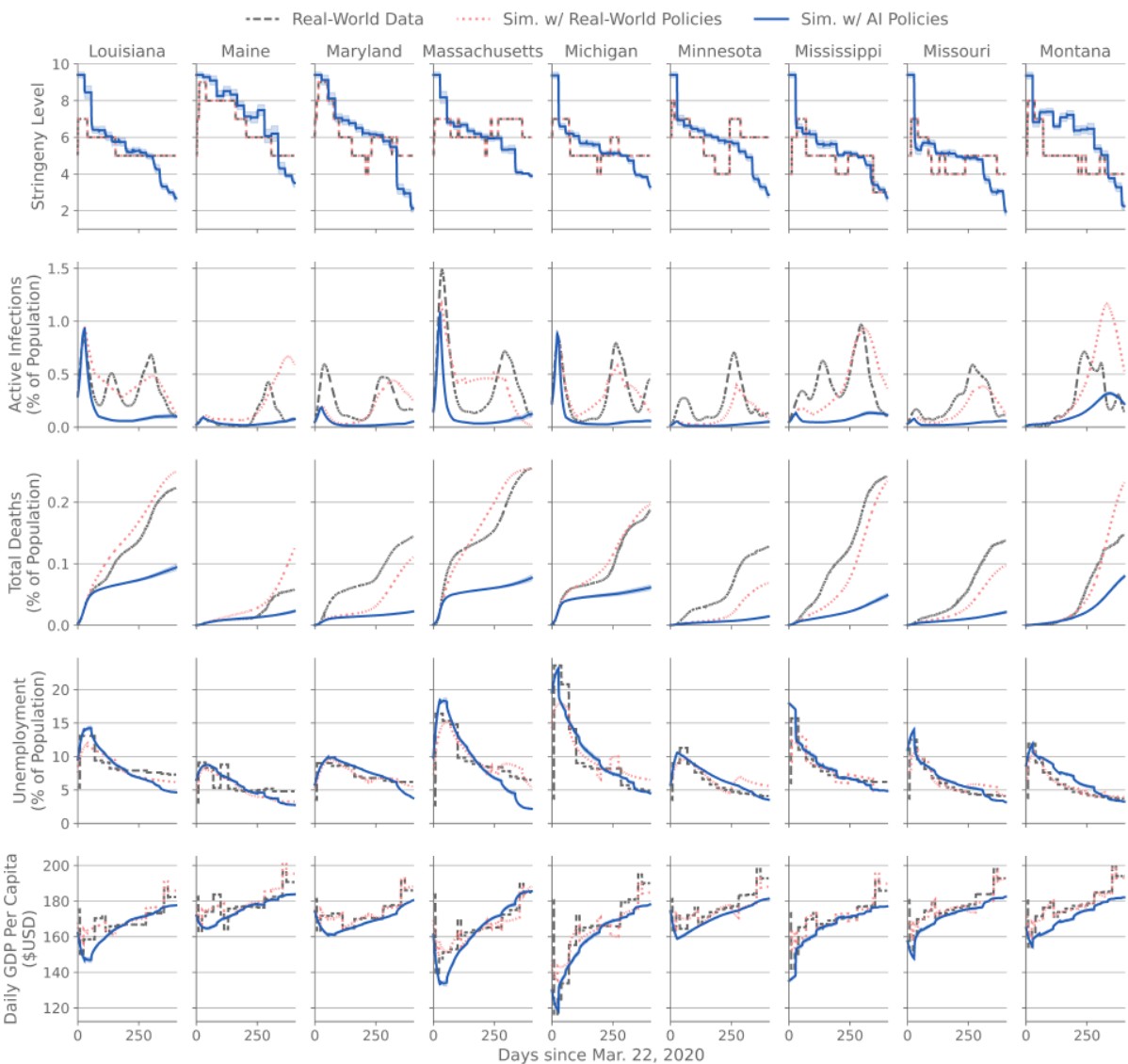

Figure 13: **Real-world vs simulation outcome at the state level (Louisiana - Montana).** Conventions are the same as Figure 6 in Section 5 of the main text.

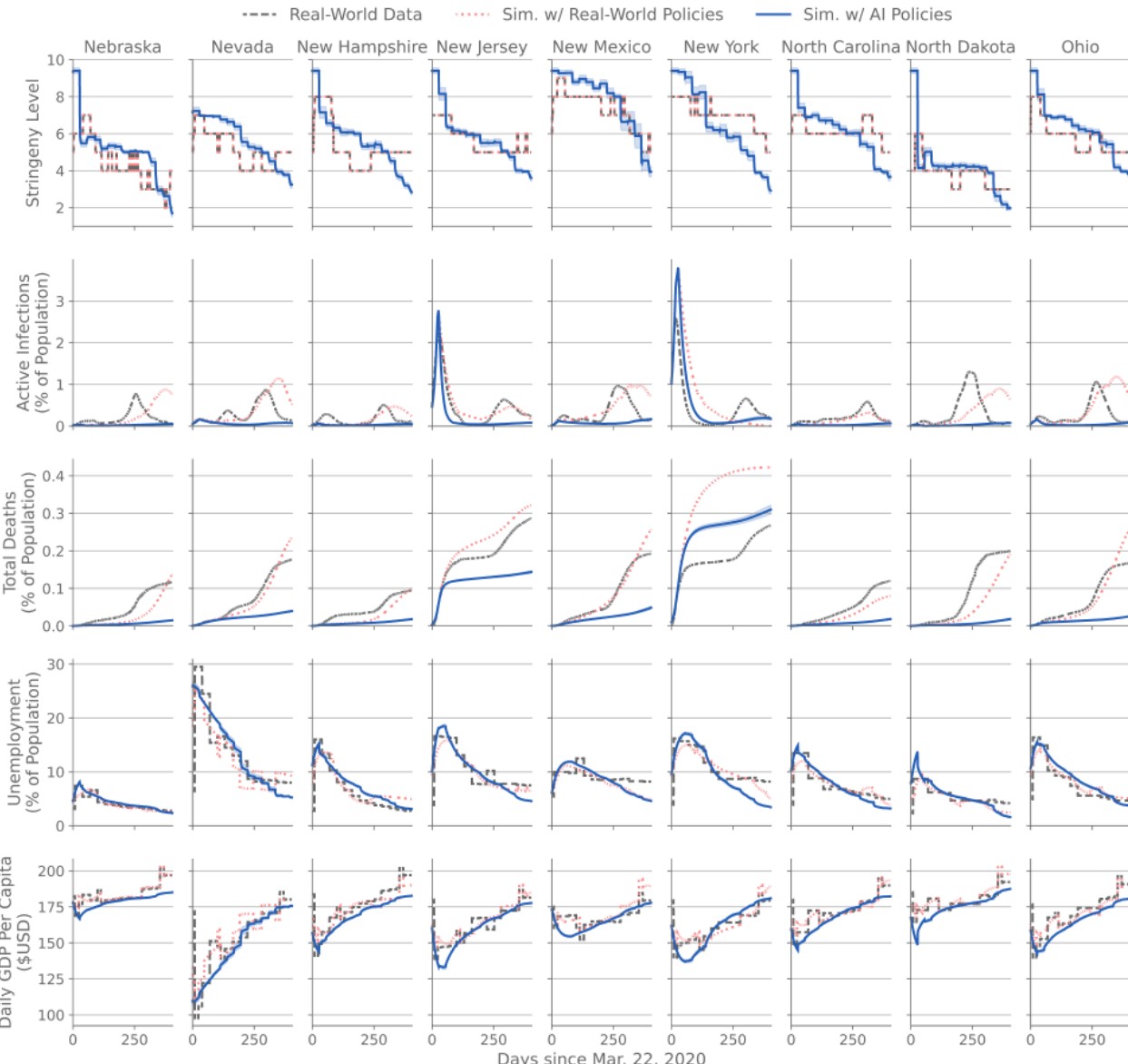

Figure 14: **Real-world vs simulation outcome at the state level (Nebraska - Ohio).** Conventions are the same as Figure 6 in Section 5 of the main text.

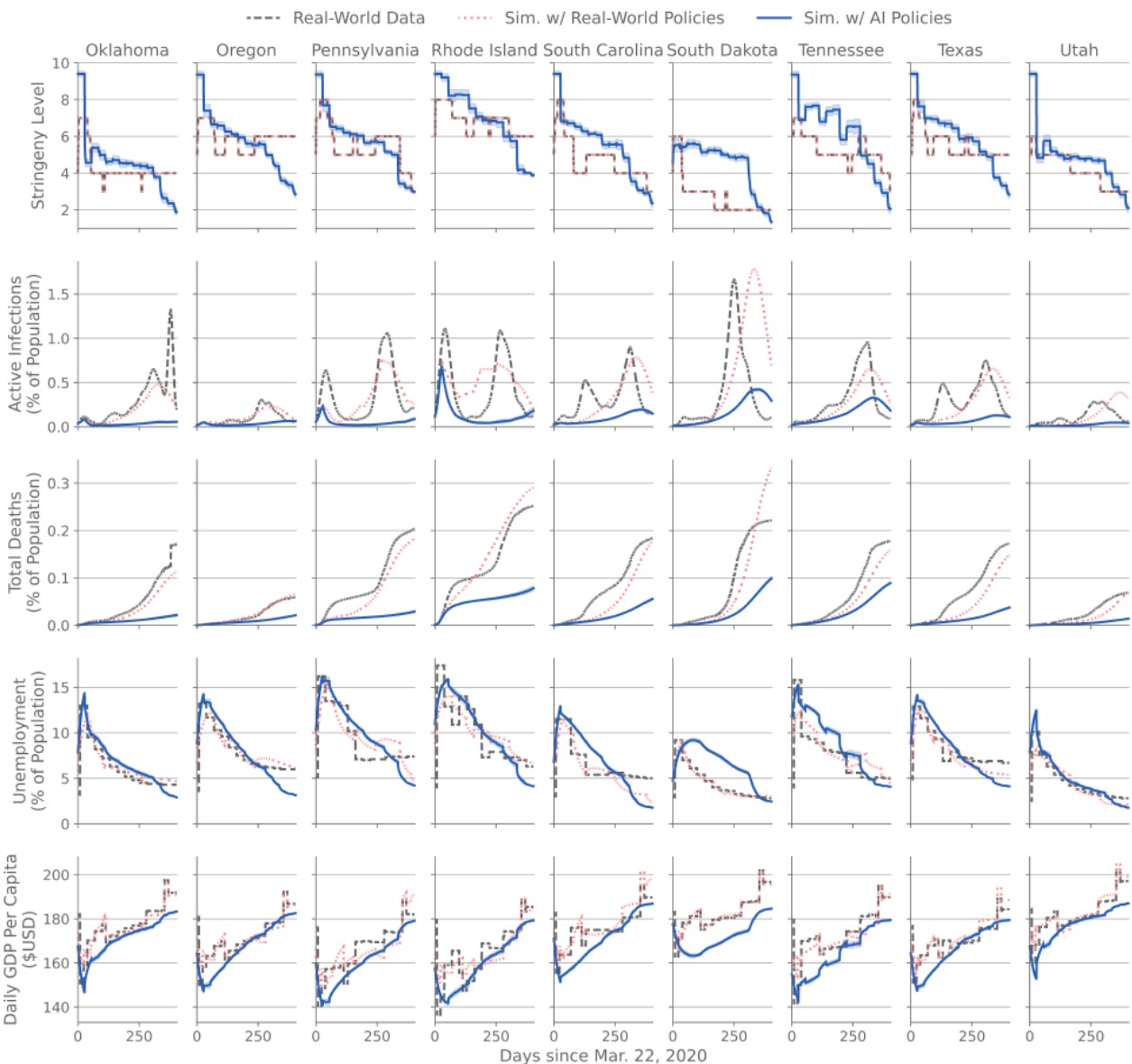

Figure 15: **Real-world vs simulation outcome at the state level (Oklahoma - Utah).** Conventions are the same as Figure 6 in Section 5 of the main text.

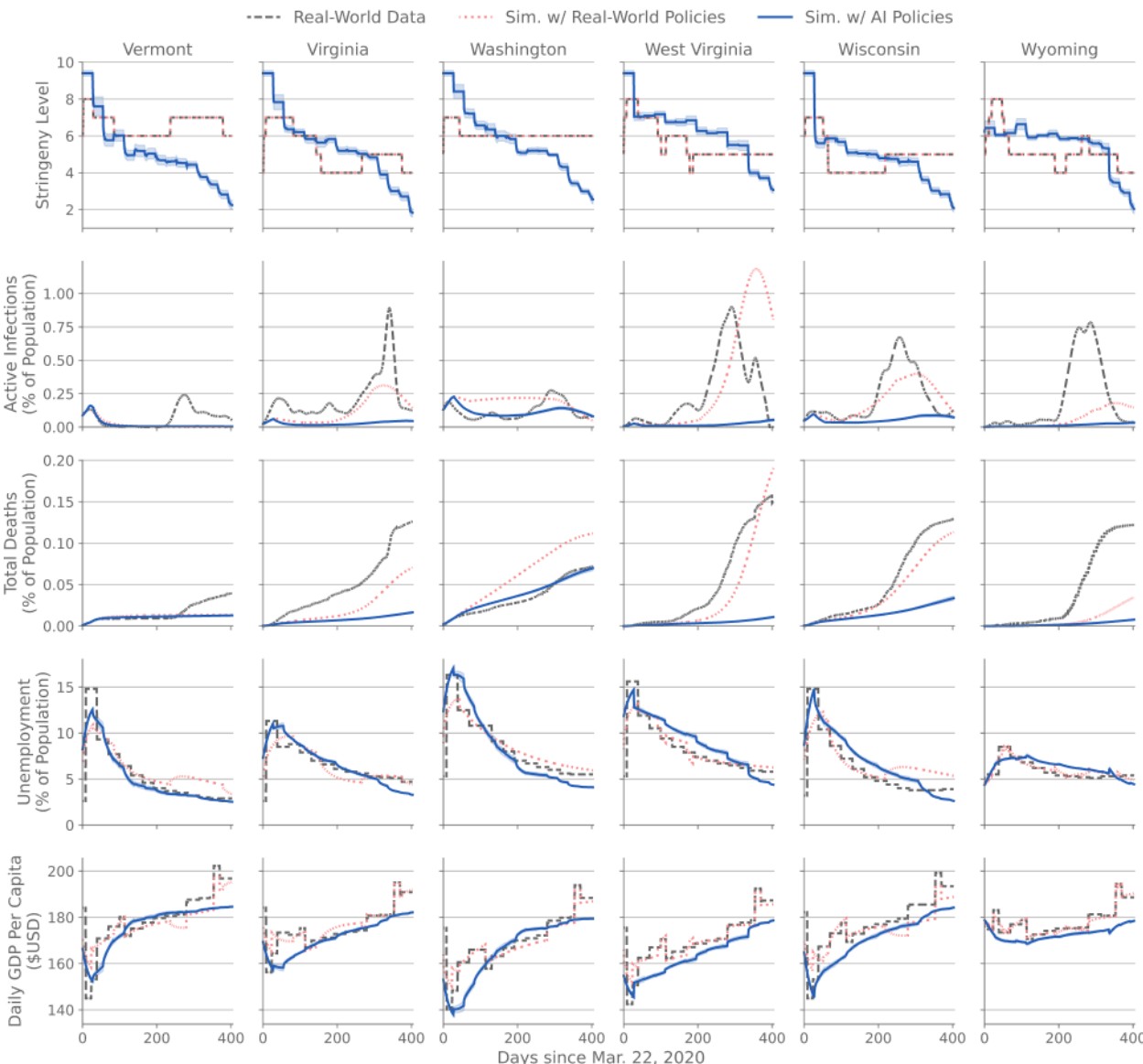

Figure 16: **Real-world vs simulation outcome at the state level (Vermont - Wyoming).** Conventions are the same as Figure 6 in Section 5 of the main text.

| Parameter | Symbol | Value |
|---|---|---|
| Episode length | $T$ | 540 |
| Number of (non-planner) agents | $N$ | 51 |
| Time | $t$ | $1, \ldots, T$ |
| Agent indices | $i$ | $1, \ldots, N$ |
| Planner index | $p$ | |
| Agent policy | $\pi_i$ | |
| Planner policy | $\pi_p$ | |
| Model weights | $\theta, \phi$ | |
| State | $s$ | |
| Observation | $o$ | |
| Action | $a$ | |
| Reward | $r$ | |
| Discount factor | $\gamma$ | |
| State-transition, world dynamics | $\mathcal{T}$ | |

Table 2: Reinforcement learning variables and indices.

