# OpenReview forum: "Effective, Explainable, and Robust Policy Design in an Economy-SIR Model using Two-Level Reinforcement Learning"
_TMLR — Rejected by TMLR_

### Review · Reviewer_MfxA · 2022-05-12

**Summary Of Contributions:**

The authors come up with a detailed COVID simulation environment that emulates what happened in the US.

The authors then use the AI economist framework to use RL to come up with state & federal policies to optimize between health & economic outcomes. The authors use a large amount of real-world data to calibrate the model. The results (in simulation) achieve better outcomes than real-world outcomes.


**Broader Impact Concerns:**

I do not have any broader impact concerns beyond what the authors bring up in their section. The authors do an excellent job discussing the potential ethical implications of the work in their Broader Impact statement. It is comprehensive and detailed.

To an extent, I think that the authors are overly cautious. For instance, they gate the code, requiring individuals to provide a description of how they intend to use the code. I don’t think that’s necessary. Most of the bad uses would, in my opinion, be due to an institution using these models to blindly make decisions. Gating the code makes it less likely that any issues would be discovered (e.g. if a researcher has to justify getting access, they are less likely to get access than if the code were publicly available).

I quite like their “simulation card” idea; I think that is an excellent way to discuss the assumptions & limitations of a system.

**Requested Changes:**

1. I would like to see some sort of baseline. Does the proposed framework outperform other techniques?
2. I would like to see an analysis of how results vary as the social welfare function changes. What if the social welfare function was purely centered around economic output?
3. I would like to see an analysis of different variants. What if the mortality rate doubled? Or what if the recovery rate halved?
4. How does changing the timescale of the model decisions affect output?
5. I would make the Broader Impact section more concise. The risks are, basically, that policymakers would use the model inappropriately. I think that this could be expressed more concisely.
6. More generally, the writing of the paper could be made more concise. The paper tends to be very detailed and verbose. I think that it would benefit from copy-editing to make the paper shorter.
7. What exactly are the real-world policies? Please go into more detail as to how the policies are defined.


**Strengths And Weaknesses:**

A strength of this paper is the excellence in the experiments. The RL experiments are done well, with a level of empirical rigor that is admirable. The use of real data is excellent. If you buy the assumptions the authors make, then their experiments are strong.

However, I don't think the assumptions are at all realistic. The core assumption here is that optimizing economic policies is a complex multi-agent challenge that RL can be used to solve, as “existing analytical and computation methods do not scale to this setting.” I don’t think this is quite correct. I could imagine some sort of representative agent model which solves a simplified version of this problem. Say, for instance, that we have a “never close” or a “never open” baseline. How would that do? What if we have a few other simplified models? Without any baselines, it is difficult for me to ascertain the benefit of this approach.

If the goal is to use AI to guide policy design and improve social welfare, we should compare it to existing methods. I am not convinced that RL does better here than any other existing techniques.

My fundamental critique here is that I don’t think that the problem the authors solve- deciding how to choose between optimizing for economic outcomes vs health outcomes- is captured in their data. The key problem is, in my mind, that politicians in the United States govern only through the consent of the governed. Politicians were, and are, severely constrained by the willingness of the populace to submit to measures. Politicians have strong incentives to take actions that will lead them to be reelected. A better baseline might be to compare against the actions that were advocated for by public health experts, which would presumably be less constrained by political priorities.

The authors make a strong claim in their paper; namely, that:

“Compared to real-world policies, RL policies (blue lines) impose comparatively higher stringency at the start of the outbreak but reduce stringency more rapidly. Similarly, RL policies result in more unemployment early on but recover towards pre-pandemic levels more quickly. Overall, however, unemployment under RL policies is higher on average during the analysis window. Moreover, RL policies result in considerably fewer COVID-19 deaths in this simulation.”

I do not think that the authors have sufficient evidence to support this claim.

Another critique is that the authors model each agent as maximizing their social welfare, which is a weighted sum of a health index H and an economic index E. I think this somewhat unnatural. As an individual, I don’t have health preferences per se, but rather, I have preferences over my future happiness. To a certain granularity, this can be approximated by discounted future utility, which we choose to model here by economic output. As such, I think a model where the planner is seeking to maximize the discounted output of the economy as a whole (over a long period of time) would be more natural.

Ultimately, the authors solve a simplified version of the economy that appears tractable with classic techniques to me. I would appreciate more discussion from the authors as to why existing methods from the economic literature are not appropriate here.

The model makes unrealistic assumptions. For instance, it only models the outbreak of a single variant of COVID-19, and assumes that individuals can become infected and then recover. This is not the case. Given that RL removes a lot of constraints on analytical feasibility, an advantage of using reinforcement learning to solve a scenario is that one can layer on constraints. For instance, one could introduce multiple variants, in line with the real world, and give individuals varying probabilities to be reinfected. Using a simplified model is necessary to make it easier to solve analytically; this is not needed with RL techniques, as it is much easier for RL agents to solve complicated scenarios.

In short, the paper seems to make the claim that the fundamental problem here was one of poor decision making: real-world policymakers lacked the insight (or perhaps tools) to make proper policy decisions. This seems unlikely to me; it seems more likely that there is some constraint here, such as political feasibility, that is not being modeled.

I think that reinforcement learning has a role to play in policy decision-making, e.g. in simulating complex situations, but this paper has not convinced me that the authors have accomplished this.

---

### Review · Reviewer_j1R4 · 2022-05-16

**Summary Of Contributions:**

This paper presents an interesting case study that applies the planning algorithm for multi-agent reinforcement learning (MARL) in a synthetic model representing the impact of the COVID19 pandemic on the US economy. The epidemiology model includes the effects of vaccines, recovery, and death of individuals. Meanwhile, the economic output is modeled via unemployment and direct payments from the federal government. The authors fit the models using US COVID-19 data collected from observational studies. In this unemployment-vaccination-SIR simulation environment, the authors found that log-linear RL policies achieve significantly higher social welfare than real-world policies executed in simulation. Across states, well-performing agents use harder and faster response policies to lower deaths, while leading to similar levels of unemployment after about one year.

**Broader Impact Concerns:**

The authors have sufficiently addressed the broader impact of the proposed method.

**Requested Changes:**

To follow up on my previous points, while I appreciated the clear statement of parametric assumptions, I would like to see a discussion about the causal implications of these parametric assumptions. In the most general case, the observational distribution underdetermines the interventional distribution, especially when the unobserved confounding is present. On the other hand, reinforcement learning studies the optimal policy that the agent could follow to act (i.e., intervene) in the environment. Since the simulated model is calibrated based on the observational data, additional identification analysis is required to make comments about the effects of learned RL policies. I would suggest the authors include a discussion about the identification analysis of target effects, possibly based on classic conditions including ignorability (or equivalently, backdoor criterion).

Also, based on the current form of the paper, I would suggest removing "explainable, and robust" in the title, since the discussion on these two topics appears limited. The current title could be a bit confusing.

**Strengths And Weaknesses:**

I appreciate the idea of combining reinforcement learning algorithms with economic models of real-life applications, e.g., the COVID-19 relief program. The paper is clearly written. The authors have explicitly explained parametric assumptions behind the epidemiology and economy models. The learned RL policy demonstrates desirable properties in the simulated environment, e.g., improving public health and economic metrics compared to real-world outcomes. This RL policy takes the form of an exponential linear family distribution. This allows one to obtain a reasonable interpretation of its behavior via feature attribution.

On the other hand, I found some of the contributions of this paper a bit overstated. For instance, the title says "effective, explainable, and robust policy design in an economy-SIR model". I think the paper has demonstrated that in the simulation model, the obtained RL policy is effective, outperforming real-life data. However, as for the "explainable", It appears to me that the explainability of the learned policy comes from the log-linear form of the distribution. The interpretation is obtained by applying the feature attribution from (Ribeiro et al., 2016). No novel explanation method is introduced.

Also, I have some concerns with regard to the robustness claim of the learned policies. Since the simulated model is calibrated on the observational data, it only represents one possible instance that generates the US COVID19 data. However, the actual data-generating model might be significantly different from the simulated one. Consequently, deploying the learned RL policy in real-world applications could lead to sub-optimal performance. The authors perform a robust analysis by perturbing the unemployment rate and COVID transmission-rate response parameters. Still, the actual model may not follow the assumed parametric form. Additional model validation analysis should be included when discussing the robustness of the learned RL policy.

---

### Review · Reviewer_m5K9 · 2022-05-27

**Summary Of Contributions:**

The paper studies policymaking for the covid-19 pandemic through an AI economist framework. In this setting, the federal government provides subsidies to the states to account for slower economic growth due to the lockdown and infection spread. The states then use these subsidies to set (possibly state-dependent) policy stringency levels within their own states. Given the actions of both the federal government and the states, the infection level and economic health signals are observed within each state. This interaction then repeats a finite number of times.

The authors use multi-agent reinforcement learning to derive effective and explainable policies for both the federal government and the states. The authors compare stringency level and the resulting unemployment and death rates of their policy to real-world data and show that the learned policy will result in fewer deaths and lower unemployment by employing a more stringent policy towards the end.

**Broader Impact Concerns:**

I enjoyed reading about the broader impact in section 8 and in general really appreciate all the points raised in the discussion.

**Requested Changes:**

I would like the authors to address these questions or add them to a shortcoming section.

Assumptions:

--The assumption that the recovered people do not become susceptible again does not clearly hold in practice. There have been many people who have been infected more than once. Can this assumption be relaxed (possibly by using simulation to derive the SIR values instead of the closed-form)? Is this the reason for the discrepancy in the left and middle panels of Figure 1? (I would also know why the simulated w AI policy mirror the real-world data for the most part but not towards the end?)

--The SIR model does not capture infection spread between the states. Does this assumption technically convert the dynamics into n independent 2-agent settings? As far as I can see the utilities of the agents are not tied in any other way to each other besides possibly the spread of the infection.

Exposition:
--Section 9 contains the model and assumptions. I would recommend moving this section to the earlier parts of the paper. It probably does not need to be as detailed (for estimating the parameter) but I would like the setting and assumptions to be clearly stated before presenting the results.

Clarifications/Quesitions:

--The paper claims the parameters are estimated from the 2020 data and the results are reported with respect to the 2021 data. Why do the axes in Figure 3 goes to 400 and not 365? Is this training performance? It also says days since March 22, 2020, which is confusing.

--The SIR model has a closed-form solution. However, I am unfamiliar with the SIR-vaccine model that is used here. Do the closed-form solutions carry over or new tools are needed? This should be clarified.

--There are many parameters in the model and it is not clear how these values are estimated from the text (in particular, the alpha values).

--The details of training the RL agents are missing as far as I can tell.

--How did you come up with the daily subsidy of 0-55 for each person? Why do you test in increments of 20? Can the discrepancy between your result with real-world data be explained by the fact that this budget is not distributed uniformly in the real world but you assume a uniform distribution?

--Why Figure 6 does not exhibit monotonicity? i.e. fixing m_{1,N}, the trend should be either increasing or decreasing as m_p increases. This is not the case.

--There are no baselines for the experiments. One time-step version of this game is just a Stackelberg game. Can game-theoretic approaches be used as baselines? If not, a clear justification is needed.

--There has been a plethora of work studying policymaking during covid. I feel like the related work fell short of doing a comprehensive comparison with this previous work and highlighting the novelty/significance of the paper.

**Strengths And Weaknesses:**

Strengths:

-- The paper studies pandemic policymaking. This has merit given the importance of the problem and the difficulty of decision-making in this setting in practice/real-time.

-- The results are interesting (though not very surprising).

-- The paper is clearly written (especially in the first 6-7 pages) though some details are vague later in the paper. See below for more details.

Weaknesses:

-- The paper utilizes real data for training which might not be available when making decisions in real-time.

--The paper does not provide any technical novelty (though apparently, this is not a significant criterion for TMLR).

--The exposition of the paper can be improved. Many of the technical details about the setting and assumptions are defined in Section 9. This limits the readability of the paper.

--There are some strong assumptions e.g. the lack of spread of infection between the states and the absence of reinfection. See below for more details.

---

### Decision · Action_Editors · 2022-07-20

**Recommendation:** Reject

**Comment:**

Unfortunately, we cannot accept this paper in its current form.  While the authors revisions have certainly addressed some of the concerns of the reviewers, there are still some fundamental concerns around (potential overstatement of) claims of the paper and whether they are justified given the assumptions of the work even if pragmatically justified.

Notably, there are limitations of the model (e.g., lack of reinfection or additional variants, infection between states).  The one significant comparator of real-world outcomes was not under the same objective (e.g., political feasibility was clearly a major concern in the decisions made).  And, the RL system was optimized for a model fitted to data that wasn't known by decision-makers in real-time.  All of this makes it unclear about what can be claimed about the algorithm "out-performing" anything.

In all, this makes me wonder what is the targeted message of this work.  Is it that health authorities should consider using the author's approach in adopting policy rather than their current methods?  Is it that politicians should, even if they are optimizing a different objective?  Is it that the AI Economist Framework is better than other modelling approaches (e.g., game theoretic optimization) to constructing policy?  Being very clear on this and then focusing claims on what can be justified with is in mind would go a long way.